# REASONING AS META-LEARNING: AN OPTIMIZATION PERSPECTIVE TO DECIPHER LONG COT REASONING IN LLMS

## ABSTRACT

We propose a novel framework RAML for interpreting the reasoning capabilities of large language models (LLMs) through the perspective of meta-learning. By conceptualizing reasoning trajectories as pseudo-gradient descent updates to the LLM's parameters, we identify parallels between LLM reasoning and various meta-learning paradigms. We formalize the training process for reasoning tasks as a meta-learning setup, with each question treated as an individual task, and reasoning trajectories serving as the inner loop optimization for adapting model parameters. Once trained on a diverse set of questions, the LLM develops fundamental reasoning capabilities that can generalize to previously unseen questions. Extensive empirical evaluations substantiate the strong connection between LLM reasoning and meta-learning. We further explore the potential of the proposed RAML to advance LLM reasoning and provide valuable insights. Our work deepens the understanding of LLM reasoning processes and provides actionable insights for enhancing these models through established meta-learning techniques.

## 1 INTRODUCTION

Recent advancements in large language models (LLMs) (Dubey et al., 2024; Yang et al., 2024a; OpenAI, 2023; DeepSeek-AI et al., 2024) have significantly improved their capacity to perform complex reasoning tasks. Current LLMs often utilize chain-of-thought (CoT) reasoning (Wei et al., 2022; Chen et al., 2025b) (i.e., intermediate reasoning trajectories), to facilitate systematic problem-solving through coherent, step-by-step logical deductions. Among them, state-of-the-art LLMs, such as OpenAI-o-series (OpenAI, 2024b;a), DeepSeek-R1 (DeepSeek-AI et al., 2025), Kimi-k1.5 (Team et al., 2025), Qwen3 (Yang et al., 2025a), and Gemini-2.5-Pro (Deepmind, 2025), exhibit exceptional proficiency in addressing intricate mathematical and programming challenges. These models employ long reasoning trajectories, characterized by an iterative and detailed process of exploration and reflection, to enhance test-time scaling capabilities (Li, 2025; Teng et al., 2025; Shah et al., 2025). This iterative approach has driven significant progress in complex reasoning while motivating the studies to illuminate the potential of supervised fine-tuning (SFT) and reinforcement learning (RL) methods to refine the learning and application of extended reasoning processes (Qin et al., 2024; Min et al., 2024).

Despite significant advancements, ***comprehending and interpreting how LLMs achieve prominent reasoning capabilities through reasoning trajectories*** remains crucial for further enhancement and generalization (Jiang et al., 2020; Feng et al., 2023). The opaque nature of LLMs' internal mechanisms hinders efforts to comprehend their operations (Shi et al., 2025). Recent studies (Merrill et al., 2022; Chiang et al., 2023; Giannou et al., 2023; Liu et al., 2023) have explored the representational power of reasoning trajectories, showing that LLMs equipped with these trajectories can solve complex problems. Other research (Gatmiry et al., 2024; Huang et al., 2025) demonstrates that reasoning trajectories can effectively describe complex learning algorithms. Nevertheless, there is a notable gap in research exploring the fundamental role of reasoning trajectories in LLM reasoning and connecting diverse training approaches to enhance these capabilities. To address this, we propose **RAML** (Reasoning as Meta-Learning), a methodology that analyzes LLM reasoning through a meta-learning perspective (Schmidhuber, 1987; Andrychowicz et al., 2016; Ravi & Larochelle, 2017; Finn et al., 2017; Hospedales et al., 2022). We conceptualize reasoning trajectories as pseudo-gradient descent

updates to model parameters, leveraging established meta-learning methodologies, such as Model-Agnostic Meta-Learning (MAML) (Finn et al., 2017) and Learn to Optimize (L2O) (Andrychowicz et al., 2016), to enhance both the understanding and optimization of LLM reasoning.

To be more specific, RAML frames the training regimen for reasoning tasks as a meta-learning framework, wherein each question constitutes a distinct task, reasoning trajectories serve as inner-loop optimization for parameter adaptation, and answers act as the query set to optimize LLMs. In the context of RAML, the training process optimizes the LLM to develop generalized reasoning abilities, identifying an effective *meta-initialization* that enables efficient parameter adaptation through reasoning trajectories to produce accurate responses. This approach provides a theoretical foundation for analyzing LLM reasoning capabilities and training, while facilitating the application of meta-learning insights to advance LLM reasoning research.

RAML framework is complemented by comprehensive experiments and analysis involving both models trained from the base LLMs and publicly available models. Drawing on meta-learning studies (Liu et al., 2020; Triantafillou et al., 2020; 2021; Agarwal et al., 2021; Lee et al., 2019; Collins et al., 2022), we conduct experiments to explore key factors influencing LLM reasoning by framing them within a meta-learning perspective. These experiments further confirm a strong connection between trajectory-based LLM reasoning and meta-learning principles. Furthermore, we have demonstrated that RAML has the potential to advance the development of LLM reasoning. We propose potential strategies for enhancing LLM reasoning capabilities based on RAML and validate their effectiveness in improving reasoning performance. Our contributions are summarized as follows: ❶ To elucidate the reasoning processes of LLMs, we introduce RAML, an interpretation methodology for LLM reasoning from a meta-learning perspective, supported by a comprehensive theoretical analysis; ❷ We provide empirical evidence and detailed analysis, demonstrating a strong correspondence between LLM reasoning and meta-learning principles; ❸ We contextualize recent advances in LLM reasoning within our framework, offering comprehensions into their success; ❹ We propose simple methods and present meaningful insights to advance LLM reasoning, building on the existing meta-learning research and analysis addressed in our work.

## 2 RAML: INTERPRETING LLM REASONING AS META-LEARNING

In this section, we elucidate the interpretation methodology for the large language model (LLM) reasoning from a meta-learning (Schmidhuber, 1987; Andrychowicz et al., 2016; Ravi & Larochelle, 2017; Finn et al., 2017; Hospedales et al., 2022) perspective, i.e., RAML. First, we conceptualize the reasoning trajectories as a *pseudo gradient update* to the parameters of the LLM (§ 2.2) and subsequently develop a meta-learning framework to model the training process for the reasoning task (§ 2.3). As a supplement, we establish connections between various training techniques and our proposed definition in § E. The notations used in this section are listed in § A.

### 2.1 SETUP

In this paper, we represent the large language model (LLM) as $\mathcal{M}_\theta$, where $\theta$ signifies the parameters of the LLM. We focus on a specific reasoning task that involves a set of questions denoted as $\mathcal{Q} = \{q_i\}_{i \in [1,n]}$ and its corresponding answers $\mathcal{A} = \{a_i\}_{i \in [1,n]}$. Typically, the LLM is prompted to generate the answer $a$ based on the instruction $I$ and the question $q_i$:

$$\mathrm{d}\left(\mathcal{M}_\theta(I; q_i)\right) \to a_i, \tag{1}$$

where $\mathrm{d}$ denotes the autoregressive decoding mechanism (Vaswani et al., 2017; Radford et al., 2018), which is specifically defined as follows:

$$p_\theta(a_i) = \prod_{0 \le j < |a_i|} \mathrm{Softmax}\left(\boldsymbol{E}_O^T \cdot \mathcal{M}_\theta\left(I, q_i, t, a_i^0, ..., a_i^{j-1}\right)\right)\left[a_i^j\right]. \tag{2}$$

Here, $\{a_i^0, \ldots, a_i^{|a_i|}\}$ denotes the token set of the answer $a$, $t$ represents the possible intermediate reasoning trajectories, and $\boldsymbol{E}_O$ indicates the output embeddings of the entire token set (i.e., vocabulary). Intuitively, $\mathcal{M}_\theta\left(I, q_i, t, a_i^0, \ldots, a_i^{j-1}\right)$ represents the *activation* determined by the parameters $\theta$ and the inputs, while the predicted probability is computed through the inner product between the

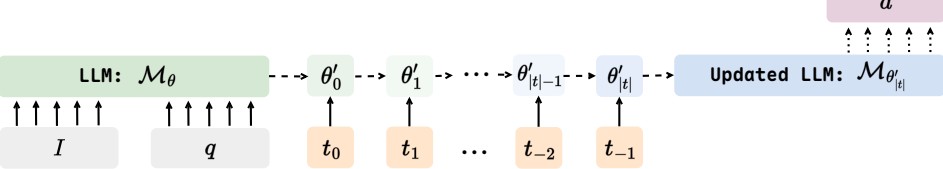

Figure 1: Illustration of the reasoning trajectory ($t$) as the optimization of the LLM parameters $\theta$.

output embeddings and the activation. The activation, representing the output at the final position, is computed iteratively through the self-attention and feed-forward layers of the LLM. During training, the LLM $\mathcal{M}$ is optimized to deliver accurate answers to each question:

$$\mathcal{O} = \max_{(q_i, a_i) \in \mathcal{Q} \times \mathcal{A}} \sum s(a_i', a_i), \tag{3}$$

where $a_i'$ means the predicted answer for $q_i$, and $s(a', a)$ indicates the plausibility of $a'$ w.r.t. $a$, which also defines a *loss landscape*.

## 2.2 REASONING TRAJECTORIES AS PARAMETER UPDATE

Recent works (Brown et al., 2020; Wei et al., 2022; OpenAI, 2024b; DeepSeek-AI et al., 2025) demonstrate that incorporating intermediate reasoning steps can significantly enhance the capacity of large language models to execute complex reasoning tasks. Moreover, some studies (Dai et al., 2023; Bai et al., 2023; Giannou et al., 2023; Gatmiry et al., 2024; Fu et al., 2024; Huang et al., 2025) theoretically demonstrate that models based on the transformer architecture can learn to perform iterative algorithms like multi-step GD with the enhancement of CoT (which we called reasoning trajectories in this paper). However, these studies primarily focus on explicit numerical optimization problems, such as linear regression, and demonstrate that LLMs can learn optimization algorithms like multi-step GD in the reasoning trajectories to solve the problem. In contrast, we conceptualize *the reasoning trajectories of an LLM $\mathcal{M}$ as a multi-step gradient descent process of the model's parameters $\theta$*, which could be formally represented by:

$$\theta_i' \leftarrow \theta_{i-1}' + \Delta\mathcal{M}_{\theta_{i-1}'}(I, q, t_{\leq i}), \quad \theta_0' = \theta, \quad 1 \leq i \leq |t|, \tag{4}$$

where $t$ denotes a reasoning trajectory, $\Delta\mathcal{M}_{\theta_{i-1}'}(I, q, t_{\leq i}) = -\eta\nabla_{\theta_{i-1}'}\mathcal{L}_q(\theta_{i-1}')$ represents the *pseudo gradient update* associated with the reasoning trajectory $t_{\leq i}$, and $\theta_{|t|}'$ signifies the updated parameters of the LLM in response to the instruction $I$, the query $q$, and the reasoning trajectory $t$.

In summary, we conceptualize each question $q_i$ as a sophisticated optimization task, with the LLM $\mathcal{M}$ being optimized to produce the corresponding answer $a_i$. Prior to generating the final answer, the LLM is guided by an intermediate reasoning trajectory, which serves as a parameter update mechanism. The overall process is illustrated in Figure 1.

**Pseudo Gradient Update.** Without loss of generality, we consider a classic transformer model (Vaswani et al., 2017) comprising a single self-attention layer and a two-layer feed-forward network while disregarding normalization layers and other components. When using $l = \{I, q\}$ as input, its activation can be expressed as follows:

$$\boldsymbol{W}_2^T \left( \sigma \left( \boldsymbol{W}_1^T \left( \text{Softmax} \left( \boldsymbol{E}_{l,-1} \boldsymbol{W}_q \boldsymbol{W}_k^T \boldsymbol{E}_{l,:}^T \right) \boldsymbol{E}_{l,:} \boldsymbol{W}_v \right) + b_1 \right) \right) + b_2, \tag{5}$$

where $\boldsymbol{E}_{l,:}$ indicates the input embeddings of the whole sequence $l$ and $\boldsymbol{E}_{l,-1}$ indicates the input embeddings of the last position of $l$. In this context, the parameters $\theta$ refers to $\{\boldsymbol{W}_q, \boldsymbol{W}_k, \boldsymbol{W}_v, \boldsymbol{W}_1, \boldsymbol{W}_2, b_1, b_2\}$. Then, given a reasoning trajectory $t$, when attending to the first token $t^0$ of $t$ activation is changed to:

$$\boldsymbol{W}_2^T \left( \sigma \left( \boldsymbol{W}_1^T \left( \text{Softmax} \left( \boldsymbol{E}_{t,0} \boldsymbol{W}_q \boldsymbol{W}_k^T \begin{bmatrix} \boldsymbol{E}_{l,:} \\ \boldsymbol{E}_{t,0} \end{bmatrix}^T \right) \begin{bmatrix} \boldsymbol{E}_{l,:} \\ \boldsymbol{E}_{t,0} \end{bmatrix} \boldsymbol{W}_v \right) + b_1 \right) \right) + b_2. \tag{6}$$

**Proposition 2.1** (One-Step Pseudo Gradient Update). *There exists a set of parameters, denoted as $\theta_t'$, which includes $\{\boldsymbol{W}_q', \boldsymbol{W}_k', \boldsymbol{W}_v', \boldsymbol{W}_1, \boldsymbol{W}_2, b_1', b_2'\}$, allowing Equation (6) to be expressed in the following form:*

$$\boldsymbol{W}_2'^T \left( \sigma \left( \boldsymbol{W}_1'^T \left( \text{Softmax} \left( \boldsymbol{E}_{l,-1:} \boldsymbol{W}_q' \boldsymbol{W}_k'^T \boldsymbol{E}_{l,:}^T \right) \boldsymbol{E}_{l,:} \boldsymbol{W}_v' \right) + b_1' \right) \right) + b_2', \tag{7}$$

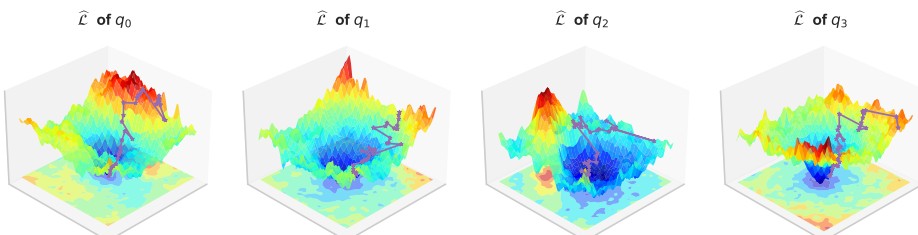

Figure 2: Landscape of the plausibility regarding LLMs to generate accurate answers. We apply the methodology proposed by Li et al. (Li et al., 2018). The questions $q_0, q_1, q_2, q_3$ are selected from AIME24. Additionally, we project the trajectory of the pseudo-gradient update onto the landscape (purple line). Please refer to § G.1 for more details.

*where $\theta'_t$ represents the one-step update of $\theta$ and the increment $\Delta\mathcal{M}_\theta(I, q, t^0)$ is only associated with $\theta$, $I$, $q$, and $t^0$.*

According to Theorem 2.1 (the proof can be found in § D), as the model progressively attends to the entire reasoning trajectory, the model parameters $\theta$ are updated incrementally, a process referred to as the *pseudo gradient update*.

**Empirical Evidence.** We provide empirical evidence for the pseudo-gradient update where the model's confidence in the answer functions serves as a probe. Specifically, we calculate the negative log-probability of the answer at each position (denoted as $\widehat{\mathcal{L}}$) within the generated trajectories by appending `Final Answer\n\boxed{...answer...}` on each position. This method provides an alternative approach to observing the overall optimization objective (Equation (3)), with models becoming more optimal as $\widehat{\mathcal{L}}$ decreases. Figure 2 displays the corresponding landscape of the negative log-probability. More examples and details are provided in § I.3.

## 2.3 A META-LEARNING PERSPECTIVE ON LLM REASONING

Building upon the previous discussion, we present a meta-learning perspective, named RAML, for modeling the process and the training of the LLM reasoning capability. We consider each question $q_i$ within the question set as an independent task in the meta-learning. During training (e.g., supervised fine-tuning (Howard & Ruder, 2018; Radford et al., 2018; Devlin et al., 2019) or reinforcement learning (Mnih et al., 2013; Schulman et al., 2017; Ouyang et al., 2022; DeepSeek-AI et al., 2025)) on the question set $\{q_i\}$, the LLM $\mathcal{M}_\theta$ is prompted to solve a new question $q_i$ by following a reasoning trajectory $t$. Initially, the parameters $\theta$ are updated to $\theta'_t$ using one or more *pseudo gradient descent update* indicated by the reasoning trajectory $t$. Subsequently, the LLM is optimized by *pseudo second order gradient*, formally expressed as follows:

$$\min_\theta \sum_{q_i \in \mathcal{Q}} \sum_{t \in \mathcal{T}_i} \mathcal{L}_{q_i}\left(\mathcal{M}_{\theta'_t}\right) = \min_\theta \sum_{q_i \in \mathcal{Q}} \sum_{t \in \mathcal{T}_i} \mathcal{L}_{q_i}\left(\mathcal{M}_{\theta + \Delta\mathcal{M}_\theta(I, q, t)}\right), \tag{8}$$

where $\mathcal{T}_i$ denotes the set of reasoning trajectories corresponding to the question $q_i$ and $\theta + \Delta\mathcal{M}_\theta(I, q, t)$ indicates the multi-step update of $\theta$ as detailed in Equation (4).

Intuitively, RAML can be perceived as a variant of **Model-Agnostic Meta-Learning** (MAML, detailed in § B) (Finn et al., 2017), where the update of $\theta$ using reasoning trajectories function as the inner loop, while the final optimization of the answer decoding distribution constitutes the outer loop, as outlined in § C. In RAML, the gradient update associated with the latent *support set* is represented by the reasoning trajectories, whereas the answer denotes the *query set*. There are no explicit evaluations (i.e., loss computation and backward) during the inner loop, as the gradient update is implicitly dictated by the reasoning trajectories. Although the model weights have not been directly updated, the pseudo update enables the LLM to simulate question-specific optimization within a specific reasoning trajectory, thereby significantly enhancing the accuracy and stability of LLM reasoning. During the training process, the parameters of the LLM, denoted as $\theta$, are updated to $\theta'_{q_i, t_j}$ according to the given reasoning trajectory $t_j$ in the inner loop. In the outer loop, the parameters of the LLM are optimized using the second-order gradient ($\mathcal{L}_{q_i} \rightarrow \theta'_{q_i, t_j} \rightarrow \theta$). The LLM is optimized

Table 1: The evaluation performance of Qwen2.5-7B-Base trained using both SFT and Zero-GRPO training methods. In this context, "Qwen" refers to the abbreviation for Qwen2.5-Math-Instruct, and "Distil-Qwen" denotes DeepSeek-R1-Distill-Qwen-14B. **Green** cells indicate the best performance in each column, while **Blue** cells indicate the second-best performance.

| Techniques | Source | AIME24 | | MATH500-L5 | | LiveMathBench-Hard | |
|---|---|---|---|---|---|---|---|
| | | Pass@8 ↑ | mG-Pass@8 ↑ | Pass@8 ↑ | mG-Pass@8 ↑ | Pass@8 ↑ | mG-Pass@8 ↑ |
| SFT | Qwen | $20.34_{\pm0.62}$ | $7.43_{\pm0.58}$ | $58.42_{\pm0.85}$ | $35.65_{\pm0.74}$ | $26.77_{\pm0.91}$ | $7.43_{\pm0.53}$ |
| | Distil-Qwen | $36.69_{\pm0.88}$ | $10.29_{\pm0.67}$ | $82.98_{\pm0.95}$ | $45.79_{\pm0.79}$ | $25.15_{\pm0.64}$ | $10.46_{\pm0.81}$ |
| Zero-GRPO | - | $27.37_{\pm0.72}$ | $4.08_{\pm0.55}$ | $71.66_{\pm0.93}$ | $30.48_{\pm0.66}$ | $27.48_{\pm0.89}$ | $8.21_{\pm0.51}$ |

Table 2: Performance of Zero-GRPO model and GRPO model based on the SFT cold start.

| Techniques | AIME24 | | MATH500-L5 | | LiveMathBench-Hard | |
|---|---|---|---|---|---|---|
| | Pass@8 ↑ | mG-Pass@8 ↑ | Pass@8 ↑ | mG-Pass@8 ↑ | Pass@8 ↑ | mG-Pass@8 ↑ |
| Zero-GRPO | $27.37_{\pm0.68}$ | $4.08_{\pm0.97}$ | $71.66_{\pm0.58}$ | $30.48_{\pm0.87}$ | $27.48_{\pm0.84}$ | $8.21_{\pm0.88}$ |
| + SFT Cold Start | $35.87_{\pm1.00\uparrow31\%}$ | $11.23_{\pm0.78\uparrow175\%}$ | $82.42_{\pm0.61\uparrow15\%}$ | $44.92_{\pm0.67\uparrow47\%}$ | $42.17_{\pm0.67\uparrow53\%}$ | $18.84_{\pm0.97\uparrow129\%}$ |

to provide an effective and robust foundation for answering questions (tasks), *allowing its parameters to be easily adapted based on the reasoning trajectories associated with these questions*, thereby facilitating answer generation.

Additionally, in the standard MAML process, a few-shot support set is typically required to fine-tune the model on a new task. In the LLM reasoning scenario, this support set, comprising reasoning trajectories, is generally generated by the LLM itself. Thus, the inner loop's optimization process in RAML resembles **Learn-to-Optimize** (L2O) (Andrychowicz et al., 2016; Li & Malik, 2017a;b), which involves learning a parameterized optimizer to automate the optimization of various tasks. Specifically, during the LLM reasoning training, the LLM is trained to function as the meta-optimizer, generating an inner loop optimization path tailored to the specific question.

## 3 EMPIRICAL ANALYSIS ON LLM REASONING FROM META-LEARNING PERSPECTIVE

Building on a meta-learning perspective of LLM reasoning, this section explores key factors that influence LLM reasoning. Specifically, we study and analyze key issues of interest in the research community regarding LLM reasoning by instantiating them within the framework of meta-learning, referencing relevant research findings in this domain. We focus on the following problems: ❶ Which training strategy, SFT or RL, is more effective for LLM reasoning, and why (§ 3.2)? ❷ Why do longer reasoning trajectories enhance reasoning performance (§ 3.3)? ❸ What principles behind reasoning-efficiency methodology contribute to the trade-off between cost and performance (§ 3.3)? ❹ Does trajectory-aided reasoning generalize effectively across different domains (§ 3.4)?

### 3.1 EXPERIMENT SETUP

**Reasoning Task.** In this paper, we mainly focus on the mathematical reasoning task due to its broad applicability and prominence in the research community and we also include other reasoning tasks in § 3.4 for the study of generalization.

**Training.** To minimize the impact of the post-training, we train Qwen2.5-7B-Base (Yang et al., 2024a) from scratch and conduct experiments on it. We involve SFT for the off-policy training and (**Zero**-)GRPO (Shao et al., 2024) for the on-policy training. The training data, sourced from Open Reasoner Zero (Hu et al., 2025), initially comprised approximately 57k questions, refined to 39k through filtering (see § G.2 for details). Synthetic reasoning trajectories are generated using Qwen2.5-Math-72B-Instruct (Yang et al., 2024b) and DeepSeek-R1-Distill-Qwen-14B (DeepSeek-AI et al., 2025). Further training details are provided in § G.2.

**Evaluation.** We primarily evaluate performance using three mathematical reasoning benchmarks orthogonal to the training data: AIME24 [1], MATH500 (Lightman et al., 2024) (Level 5 questions

---
[1] https://huggingface.co/datasets/AI-MO/aimo-validation-aime

selected for greater discrimination), and LiveMathBench-Hard (Liu et al., 2024). We also include GPQA (Rein et al., 2023) and LiveCodeBench (Jain et al., 2024) to assess generalization. Model outputs are generated with a temperature of $1.0$, top-$p$ of $0.8$, top-$k$ of $50$, and a maximum output length of $16,384$ tokens. We report mG-Pass@$k$ (Liu et al., 2024) for stability and Pass@$k$ (Chen et al., 2021) for performance. Additional evaluation details are in § G.3.

## 3.2 Inner Loop Optimization *v.s.* Reasoning Trajectory Source

The inner loop optimization is crucial in meta-learning, as the results of this process significantly impact the stability and performance of the final model. In the definition of RaML, the LLM needs to learn as the inner loop optimizer, generating an optimization trajectory for each query. It is well-documented that training learned optimizers presents considerable challenges (Lan et al., 2024). In this section, we will discuss the development of inner loop optimizers for SFT and GRPO training techniques. The biggest difference between them is the source of the reasoning trajectories used for training: 1) on-policy, where the trajectories are generated by the LLM currently being updated, and 2) off-policy, where the trajectories are generated either by other LLMs or by a previously trained version of the same LLM (e.g., through reject sampling (Yuan et al., 2023; Singh et al., 2024)).

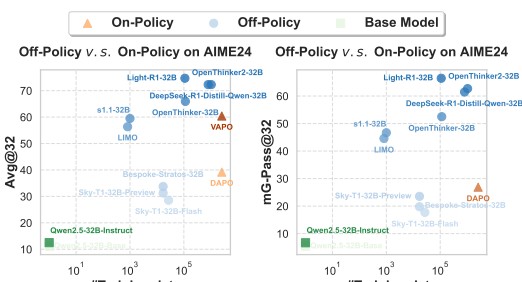

Figure 3: Performance of base models, models trained on off-policy data (SFT), and models trained on on-policy data (RL) using the AIME24 benchmark, with the $x$-axis representing the amount of training data. We generate $64$ for each question and report Pass@32 and mG-Pass@32. Details of models are provided in § G.4.

**Status Quo of SFT v.s. RL.** Recent studies (Xiong et al., 2024; Chen et al., 2025a; Chu et al., 2025a) claim the superiority of the on-policy strategy in LLM reasoning training. For example, GRPO-based DeepSeek-R1-Zero (DeepSeek-AI et al., 2025) outperforms DeepSeek-V3, which is trained on large-scale off-policy synthetic data, in mathematical reasoning tasks, scoring 71.0 compared to 39.8 on AIME24, thereby reinforcing the advantages of on-policy strategies. However, as our results in Table 1 and the evaluation results of community models in Figure 3, GRPO-trained models **do not** consistently outperform SFT-trained models for the same base LLM, consistent with findings in (DeepSeek-AI et al., 2025; Wen et al., 2025; Yang et al., 2025a).

**SFT Leads to Stable Inner Loop Optimization.** Learning to optimize frequently encounters challenges such as unstable training, easy divergence, and limited generalization. To address these issues, researchers (Prémont-Schwarz et al., 2022; Thérien et al., 2024) have suggested employing optimal optimizers as "guardian" optimizers, integrating their features to ensure convergence and stability. The training reasoning trajectories used by SFT originate from human-annotated or other advanced reasoning models. These trajectories can be viewed as guides from an *oracle optimizer*. Consequently, SFT achieves a stable and effective inner loop optimization process, leading to superior performance. However, this does not imply that reinforcement learning always has disadvantages. RL provides greater freedom to explore optimization paths and, given sufficient model capability and exploration steps, offers a higher theoretical upper limit.

**Combination of SFT and RL for Stable Inner Loop Optimization.** A straightforward idea involves training the LLM using an optimal optimizer to stabilize its performance. Subsequently, reinforcement learning can be employed to explore improved paths for inner loop optimization. As evidenced in Table 2, the RL model demonstrates substantial enhancements after supervised fine-tuning.

**Takeaway.** ❶ SFT provides stable inner loop optimization by training on trajectories from *oracle inner loop optimizer* compared with RL. ❷ Combining SFT with RL shows significant performance improvements by utilizing SFT for initializing inner loop optimization and RL for further exploration.

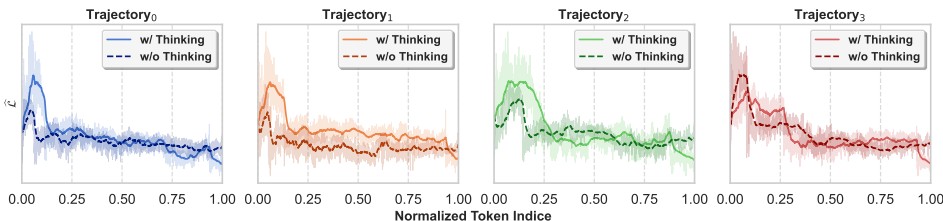

Figure 4: Illustration of QwQ's *pseudo-gradient update* for both thinking and non-thinking modes. We visualize four pairs of correct reasoning trajectories for one question in AIME24. Compared with thinking trajectories, no-thinking trajectories converge more quickly, which also easily falls into local optimal points.

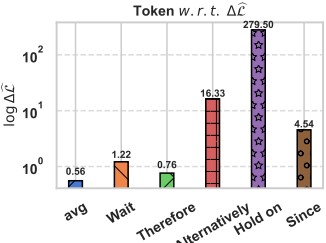

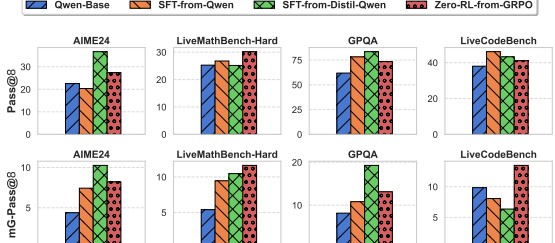

Figure 5: Illustration of the influence of reflection tokens. Reflection tokens lead to sharper objective change.

Figure 6: Evaluation results of base, SFT and GRPO models on AIME24, LiveMathBench-Hard, GPQA-Diamond, and LiveCodeBench.

### 3.3 INNER LOOP OPTIMIZATION STEPS *v.s.* REASONING TRAJECTORY TOKENS

In RAML, each token in a single reasoning trajectory corresponds to an individual optimization step, and the length of the trajectory indicates the total number of update steps. We examine these factors by integrating experimental results with related research studies.

**Long Reasoning Trajectories Lead to Superior Performance.** As shown in Table 1, models trained with longer reasoning trajectories consistently outperform those with shorter trajectories, aligning with meta-learning findings that extended inner loop updates enhance performance. This observation is consistent with the superior performance of long CoT reasoning models, such as DeepSeek-R1 (DeepSeek-AI et al., 2025), suggesting that longer trajectories increase inner update steps, thereby improving LLM reasoning capabilities.

**Different Reasoning Trajectory Tokens Represent Varying Roles of Update.** We focus on discussing two intriguing findings in LLM reasoning. First, advanced reasoning in LLMs has been observed to have an ***aha moment*** (DeepSeek-AI et al., 2025). This refers to specific *reflection tokens* that prompt LLMs to devote additional time to thinking about questions. These tokens are also utilized to implement test-time scaling (Muennighoff et al., 2025; Ma et al., 2025). Following the settings described in § 2.2, we measure the relative changes in the $\widehat{\mathcal{L}}$ value before and after each token position. The results are presented in Figure 5. We observe that reflection tokens such as "Wait" and "Alternatively" indicate a significant change in the objective. From an optimization perspective, we propose that these reflection tokens assist the model in escaping saddle points. As the model gradually approaches a stable state, these tokens provide a larger gradient, thereby expanding the exploration space to find a better parameter space. In the following part, we explore the concept of reasoning efficiency, as discussed by various researchers (Qu et al., 2025; Yang et al., 2025b; Zhang et al., 2025; Ma et al., 2025; Yang et al., 2025a). This concept involves optimizing the balance between decoding cost and performance utilizing specific segments, such as the end-of-thinking token delimiter. We hypothesize that these termination delimiters enhance convergence at the optimization level, akin to the role of *momentum* in optimization, facilitating rapid convergence of model parameters within a flatter region. However, this acceleration does not always lead to the optimal point. Also refer to the settings described in § 2.2, we append the end-of-thinking token delimiter `Therefore, after all this, I believe the answer is` following the thinking token delimiter `<think>`. Figure 4 demonstrates that trajectories using the end-of-thinking token delimiter achieve quicker

convergence, confirming our hypothesis to some extent. Since the QwQ model does not completely adapt to the no-thinking mode, we include additional experiments related to Qwen3 in § I.3. These experiments further substantiate our conclusions.

**Takeaway.** ❶ Long reasoning trajectories are analogous to performing additional steps of inner loop optimization, which improves inner loop optimization and further enhances the reasoning performance of LLMs. ❷ Different tokens serve distinct functions in the inner loop optimization process. For instance, tokens associated with reflection patterns promote the exploration of optimization paths, whereas special tokens regulating the length of reasoning in the recent Long-CoT LLMs facilitate fast-converging optimization steps.

### 3.4 TASK GENERALIZATION *v.s.* REASONING GENERALIZATION

Meta-learning models typically exhibit strong generalization across tasks with similar distributions, since the models learn generalized representations for these tasks. We investigate whether this applies to LLM reasoning, where each question is a distinct task but shares fundamental reasoning skills, suggesting a similar distribution. We analyze generalization from two perspectives: within-domain generalization (same reasoning domain) and cross-domain generalization (different reasoning domains). Training data, sourced from Open Reasoner Zero (§ 3.1), consist of mathematical problems from MATH, making AIME24 and LiveMathBench-Hard suitable for within-domain evaluation. Results in Figure 6 show improved performance for both SFT and GRPO models on these benchmarks. For cross-domain generalization, we evaluated SFT and GRPO models on GPQA (scientific reasoning) and LiveCodeBench (code reasoning). As illustrated in the right section of Figure 6, all trained models outperformed the base model on both benchmarks. Our findings align with existing research. For instance, studies have shown that large models trained on code datasets exhibit strong logical reasoning capabilities (Chen et al., 2021). Additionally, research indicates that training on mathematical and code corpora mutually enhances performance (Wang et al., 2024; Hui et al., 2024; Guo et al., 2024).

**Takeaway.** Training LLMs using trajectories facilitates the learning of shared features across diverse reasoning questions. This process, akin to meta-learning, enables the parameters of LLMs to adapt efficiently by new trajectories and demonstrate generalization to out-of-distribution questions.

## 4 ADVANCING LLM REASONING DRAWING INSPIRATION FROM META-LEARNING

In this section, we demonstrate the potential of integrating meta-learning insights to improve reasoning capabilities. We introduce two **simple yet effective** methods inspired by the studies of meta-learning to enhance the reasoning performance and efficiency of LLMs, respectively and validate their effectiveness through experiments.

**Increasing Training Reasoning Trajectories per Question.** Inspired by meta-learning insights on support set size, we propose increasing the number of training reasoning trajectories per question to enhance LLM reasoning performance. We conduct experiments using SFT and GRPO, as detailed in

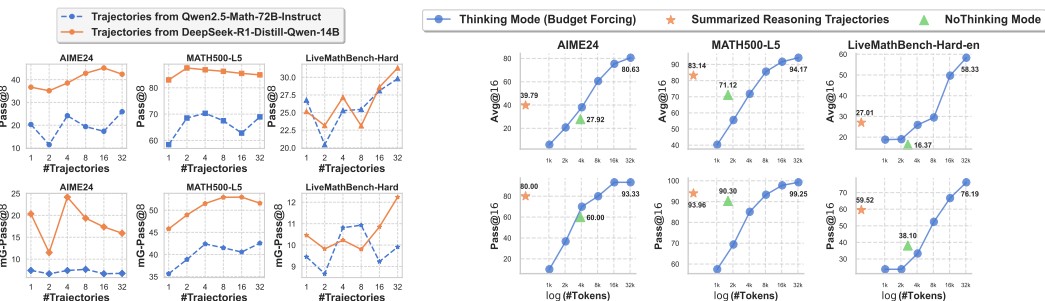

Figure 7: Performance w.r.t differ- Figure 8: Experimental results of Qwen3-32B with summa-
ent number of training trajectories per rized trajectories.
question in SFT.

§ I.1. The results, shown in Figure 7 and § I.1, demonstrate that training with multiple trajectories per question improves performance. From a meta-learning perspective, this approach is analogous to expanding the support set to enhance inner loop optimization, thereby promoting stable outer loop optimization leading to better performance.

**Incentivizing Reasoning Efficiency by Optimization Lens.** Reasoning trajectories can be viewed as optimization paths, prompting the question: *Can an optimal inner loop optimization path yield more effective reasoning trajectories?* We propose a simple method to streamline the lengthy reasoning processes of long CoT LLMs by selectively summarizing to eliminate inefficient optimization steps, as discussed in § 3.3, while evaluating the resulting performance changes. As shown in Figure 8, this approach achieves performance comparable to extensive reasoning methods and surpasses non-thinking mode performance with a reduced token count. Further details in § I.2 demonstrate that long reasoning trajectories have corresponding optimal paths that maintain performance while using fewer tokens. Developing methods to generate these optimal trajectories during decoding could enhance reasoning efficiency.

These findings confirm the feasibility of advancing LLM reasoning through a meta-learning perspective. Building on this, we propose more potential research directions in § J.

## 5 RELATED WORK

**Understanding LLMs.** The remarkable success of LLMs has spurred extensive research into their capabilities. Early studies (Yun et al., 2020a;b) explored function approximation, demonstrating that sufficiently large transformers (Vaswani et al., 2017) can universally approximate any continuous sequence-to-sequence mapping on a compact domain. Subsequent research investigated the computational power of transformers (Dehghani et al., 2019; Bhattamishra et al., 2020; Yao et al., 2021b; Hewitt et al., 2020; Weiss et al., 2021; Merrill et al., 2022; Chiang et al., 2023; Giannou et al., 2023; Liu et al., 2023). For example, Feng et al. (2023) validated the necessity of chain-of-thought (CoT) prompting for solving complex problems using circuit complexity theory. Other works (Gatmiry et al., 2024; Huang et al., 2025) demonstrate that transformers can learn to implement learning algorithms, such as gradient descent, within trajectories. Several studies (Xie et al., 2022; Akyürek et al., 2023; Dai et al., 2023; Bai et al., 2023; Olsson et al., 2022; Gatmiry et al., 2024) have focused on understanding in-context learning (ICL) (Brown et al., 2020; Dong et al., 2024), examining the role of demonstration examples.

**Meta-Learning.** Meta-learning, commonly referred to as "learning to learn", aims to enable models to enhance their learning strategies by leveraging prior experience across multiple tasks. Early research in this area (Bengio et al., 1992; Thrun & Pratt, 1998) explored methods for acquiring learning rules applicable to new tasks, with a particular emphasis on lifelong learning. These foundational efforts established the basis for creating more adaptable and flexible learning algorithms, paving the way for subsequent advancements. Recent meta-learning approaches can generally be categorized into three groups: 1) metric-based methods, which focus on learning a feature space to efficiently compare samples (Snell et al., 2017; Chen et al., 2020b; Tang et al., 2020; Zhang et al., 2023); 2) model-based methods, which utilize memory mechanisms or other structures to store and retrieve task-specific information (Weston et al., 2015; Sukhbaatar et al., 2015; Santoro et al., 2016); and 3) optimization-based methods, which refine the learning process to facilitate rapid adaptation (Finn et al., 2017; Rajeswaran et al., 2019; Ye et al., 2021).

For additional discussions on related work, please refer to § F.

## 6 CONCLUSION

This paper presents a novel perspective on LLM reasoning by integrating it with the meta-learning framework. Through theoretical analysis and empirical validation, we demonstrate that reasoning trajectories can be effectively conceptualized as pseudo-gradient updates, facilitating a deeper understanding of how LLMs perform complex reasoning tasks. Extensive experiments demonstrate the correlation between meta-learning and LLM reasoning, suggesting potential directions for advancing LLM reasoning through meta-learning principles.

ETHICS STATEMENT

This research is dedicated to upholding the highest standards of scientific integrity and abiding by ethical guidelines throughout its entire process. It centers exclusively on general research tasks, with no involvement of human subjects and no release of new datasets as part of the study. Notably, the research neither poses risks to health, safety, personal security, nor privacy, nor does it contain potentially harmful insights, methods, or applications. It also gives rise to no concerns regarding privacy, security, legal compliance, or research integrity. As a result, we expect no ethical risks or conflicts of interest to arise from this work. We are committed to maintaining the highest standards of scientific integrity and adhering to ethical guidelines throughout the research process.

REPRODUCIBILITY STATEMENT

We provide a comprehensive description of proposed RAML in § 2. For all experiments, we provide detailed implementation details, including dataset information, baseline models, and experimental settings in §§ 3.1 and G.2. All datasets used in this research are publicly available. Key code implementations are included in the supplementary materials for reference, with the complete code to be released publicly upon acceptance of the paper.

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

# Appendix

## Table of Contents

# A NOTATIONS

Table 3: Illustration of notations used in the paper.

| | |
|---|---|
| $\mathcal{M}$ | the large language model |
| $\theta$ | the parameters of the large language model |
| $q$ | the question |
| $\mathcal{Q}$ | the question set |
| $q_i$ | the $i$-th question in the question set |
| $q_i^j$ | the $j$-th token of the $i$-th question |
| $a$ | the answer |
| $\mathcal{A}$ | the answer set |
| $a_i$ | the $i$-th answer of the $i$-th question |
| $a_i^j$ | the $j$-th token of the $i$-th answer |
| $\lvert \cdot \rvert$ | the length of tokens |
| $I$ | the instruction |
| d | the autoregressive decoding mechanism |
| Softmax | the softmax function |
| $\sigma$ | the activation function |
| $t$ | the reasoning trajectory |
| $\mathcal{T}$ | the set of reasoning trajectory |
| $\boldsymbol{E}_o$ | the whole output token embedding of LLM |
| $\boldsymbol{E}_{x,:}$ | the input token embedding of the sequence $x$ |
| $\boldsymbol{E}_{x,i}$ | the input token embedding of the $i$-th token in the sequence $x$ |
| Softmax$(\cdot)[x]$ | the value of softmax vector in the entry corresponding to $x$ |
| $p(\cdot)$ | the probability distribution of one token sequence determined by the LLM |
| $\Delta \mathcal{M}_\theta(\cdot)$ | the variation of the parameter $\theta$ corresponding to the inputs |
| $\theta_i'$ | the $i$-th step updated parameters |
| $\theta_t'$ | the updated parameters $\theta$ corresponding to the reasoning trajectory $t$ |
| $\begin{bmatrix} \boldsymbol{x} \\ \boldsymbol{y} \end{bmatrix}$ | the concatenation of $\boldsymbol{x}$ and $\boldsymbol{y}$ |
| $\boldsymbol{W}_k, \boldsymbol{W}_q, \boldsymbol{W}_v$ | the parameters in self-attention layer |
| $\boldsymbol{W}_1, \boldsymbol{W}_2, b_1, b_2$ | the parameters in feed-forward network |
| $\mathcal{L}_q$ | the loss corresponding to the question |

---

**Algorithm 1:** Model-Agnostic Meta-Learning

**Input:** $p(\mathcal{T})$: distribution over tasks, $\alpha, \beta$: step size hyperparameters.

1 Randomly initialize $\theta$ ;
2 **while** *not done* **do**
3     Sample batch of tasks $\mathcal{T}_i \sim p(\mathcal{T})$ ;
4     **for** *all* $\mathcal{T}_i$ **do**
5        Evaluate $\nabla \mathcal{L}_{\mathcal{T}_i}(f_\theta)$ with respect to $K$ examples ;
6        Compute adapted parameters with gradient descent: $\theta' = \theta - \alpha \nabla_\theta \mathcal{L}_{\mathcal{T}_i}(f_\theta)$ ;

7 Update $\theta \leftarrow \theta - \beta \nabla \sum_{\mathcal{T}_i \sim p(\mathcal{T})} \mathcal{L}_{\mathcal{T}_i}(f_{\theta'})$ ;

---

## B   AN OVERVIEW OF THE MAML WORKFLOW

Algorithm 1 illustrates the algorithm of the overall workflow of Model-Agnostic Meta-Learning (MAML).

## C   PSEUDO ALGORITHM OF RAML

Algorithm 2 shows the overall flow of our proposed meta-learning perspective on LLM reasoning.

---

**Algorithm 2:** Meta-Learning Perspective on LLM Reasoning

---

**Input:** $\mathcal{M}_\theta$: LLM, $I$: instruction, $\mathcal{Q}$: question set, $\mathcal{T}_i$: reasoning trajectories for each question $i$.

1  **while** *not training done* **do**
2       Sample batch of questions $q_i$ from $\mathcal{Q}$ ;
3       **for** *all* $q_i$ **do**
4           Obtain reasoning trajectories $\mathcal{T}_i$ of each $q_i$ through training data or rollout ;
5           Update $\theta$ to $\theta'_{t_j}$ by reasoning trajectory $t_j \in \mathcal{T}_i$ refer to Equation (4) ;
6           Optimize $\theta$ through $\sum_{t_j \in \mathcal{T}_i} \mathcal{L}_{q_i}\left(\mathcal{M}_{\theta'_{t_j}}\right)$ for each $q_i$ ;

---

## D   PROOF OF THEOREM 2.1

*Proof.* Recall that our objective is to determine the set $\{W'_q, W'_k, W'_v, W'_1, W'_2, b'_1, b'_2\}$ such that:

$$W_2^T\left(\sigma\left(W_1^T\left(\text{Softmax}\left(E_{t,0}W_qW_k^T\begin{bmatrix}E_{l,:}\\E_{t,0}\end{bmatrix}^T\right)\begin{bmatrix}E_{l,:}\\E_{t,0}\end{bmatrix}W_v\right)+b_1\right)\right)+b_2 = \tag{9}$$
$$W_2'^T\left(\sigma\left(W_1'^T\left(\text{Softmax}\left(E_{l,-1:}W'_qW_k'^TE_{l,:}^T\right)E_{l,:}W'_v\right)+b'_1\right)\right)+b'_2,$$

where $E_{l,:} \in \mathbb{R}^{|l|\times d}$, $E_{l,-1}, E_{t,0} \in \mathbb{R}^{1\times d}$, $W_q, W'_q, W_k, W'_k, W_v, W'_v \in \mathbb{R}^{d\times d}$, and $W_1, W'_1, W_2, W'_2 \in \mathbb{R}^{d\times d}$.

We might as well let $W'_1, W'_2, b'_1, b'_2$ equal to $W_1, W_2, b_1, b_2$ (❶), respectively, as follows:

$$\begin{aligned}W'_1 &= W_1,\\ W'_2 &= W_2,\\ b'_1 &= b_1,\\ b'_2 &= b_2.\end{aligned} \tag{10}$$

Then, we only need to establish the following equality:

$$\text{Softmax}\left(E_{t,0}W_qW_k^T\begin{bmatrix}E_{l,:}\\E_{t,0}\end{bmatrix}^T\right)\begin{bmatrix}E_{l,:}\\E_{t,0}\end{bmatrix}W_v = \tag{11}$$
$$\text{Softmax}\left(E_{l,-1:}W'_qW_k'^TE_{l,:}^T\right)E_{l,:}W'_v.$$

For simplicity, we refine Equation (11) into several parts:

$$\begin{aligned}Q &= E_{t,0}W_q \in \mathbb{R}^{1\times d},\\ K^T &= W_k^T[E_{l,:}, E_{t,0}]^T \in \mathbb{R}^{(|l|+1)\times d},\\ V &= [E_{l,:}, E_{t,0}]W_v \in \mathbb{R}^{(|l|+1)\times d},\\ Q' &= E_{l,-1:}W'_q \in \mathbb{R}^{1\times d},\\ K'^T &= W_k'^TE_{l,:}^T \in \mathbb{R}^{|l|\times d},\\ V' &= E_{l,:}W'_v \in \mathbb{R}^{|l|\times d},\end{aligned} \tag{12}$$

where $d$ is the dimension size of embeddings.

Initially, considering the matrices $\boldsymbol{Q}$, we will demonstrate the existence of a linear transformation matrix $\boldsymbol{P} \in \mathbb{R}^{d \times d}$ such that:

$$\boldsymbol{E}_{t,0} \boldsymbol{P} = \boldsymbol{E}_{l,-1:}. \tag{13}$$

To support this assertion, we reference Theorem D.1:

**Theorem D.1.** *Let $U$ and $V$ be vector spaces, and let $\{\boldsymbol{b}_1, \boldsymbol{b}_2, \ldots, \boldsymbol{b}_n\}$ denote a basis of $U$. For $n$ vectors $\boldsymbol{v}_i \in V$, there exists a linear transformation $T : U \to V$ such that $T(\boldsymbol{b}_i) = \boldsymbol{v}_i$ for each $i = 1, 2, \ldots, n$.*

*Proof.* We begin by defining a linear transformation $T : U \to V$. Let $\boldsymbol{u}$ be a vector in $U$, expressed as $\boldsymbol{u} = u_1 \boldsymbol{b}_1 + u_2 \boldsymbol{b}_2 + \cdots + u_n \boldsymbol{b}_n$, where the set $\{\boldsymbol{b}_1, \boldsymbol{b}_2, \ldots, \boldsymbol{b}_n\}$ constitutes a basis and the coefficients $u_1, u_2, \ldots, u_n$ are determined by $\boldsymbol{u}$. The linear transformation $T$ is constructed as follows:

$$T(\boldsymbol{u}) = u_1 \boldsymbol{v_1} + u_2 \boldsymbol{v_2} + \cdots + u_n \boldsymbol{v_n}. \tag{14}$$

It is evident that this transformation $T$ satisfies $T(\boldsymbol{b}_i) = \boldsymbol{v}_i$ for each index $i$. $\square$

Based on Theorem D.1, if we define a basis which involves $\boldsymbol{E}_{t,0}$ (e.g., $\{\boldsymbol{E}_{t,0}, \boldsymbol{0}, \ldots, \boldsymbol{0}\}$) and construct a corresponding vector space, we can derive a linear transformation matrix $\boldsymbol{P}$ such that Equation (13) holds, with $\boldsymbol{P}$ being solely dependent on $\boldsymbol{E}_{t,0}$. Therefore, if we let $\boldsymbol{W}_q' = \boldsymbol{P} \boldsymbol{W}_q$ (❷), then we have $\boldsymbol{Q} = \boldsymbol{Q}'$. Consequently, we simplify Equation (11) to:

$$\text{Softmax}\left(\boldsymbol{Q} \boldsymbol{K}^T\right) \boldsymbol{V} = \text{Softmax}\left(\boldsymbol{Q} \boldsymbol{K}'^T\right) \boldsymbol{V}' \tag{15}$$

Now considering the matrix $\begin{bmatrix} \boldsymbol{E}_{l,:} \\ \boldsymbol{E}_{t,0} \end{bmatrix} \in \mathbb{R}^{(|l|+1) \times d}$ and $\boldsymbol{E}_{l,:} \in \mathbb{R}^{|l| \times d}$, we can consistently identify a vector $\boldsymbol{C} \in \mathbb{R}^{1 \times |l|}$ such that:

$$\boldsymbol{E}_{t,0} \approx \boldsymbol{C} \boldsymbol{E}_{l,:}. \tag{16}$$

We examine the existence of $\boldsymbol{C}$ in two cases: 1) if $\boldsymbol{E}_{t,0}$ lies within the span of the row vectors of $\boldsymbol{E}_{l,:}$, then $\boldsymbol{C}$ obviously exists; 2) if $\boldsymbol{E}_{t,0}$ does not lie within the span of the row vectors of $\boldsymbol{E}_{l,:}$, an approximate solution for $\boldsymbol{C}$ can be derived using various methods, such as the *least squares method* (Merriman, 1877). Then, let $\boldsymbol{M} = \begin{bmatrix} \boldsymbol{I}_l, \boldsymbol{C}^T \end{bmatrix} \in \mathbb{R}^{|l| \times (|l|+1)}$, it follows that:

$$\begin{bmatrix} \boldsymbol{E}_{l,:} \\ \boldsymbol{E}_{t,0} \end{bmatrix} \approx \boldsymbol{M}^T \boldsymbol{E}_{l,:}. \tag{17}$$

We can express this relationship mathematically as follows:

$$\begin{aligned} \text{Softmax}\left(\boldsymbol{Q} \boldsymbol{K}^T\right) \boldsymbol{V} &= \text{Softmax}\left(\boldsymbol{Q} \boldsymbol{W}_k^T \begin{bmatrix} \boldsymbol{E}_{l,:} \\ \boldsymbol{E}_{t,0} \end{bmatrix}^T\right) \begin{bmatrix} \boldsymbol{E}_{l,:} \\ \boldsymbol{E}_{t,0} \end{bmatrix} \boldsymbol{W}_v \\ &\approx \text{Softmax}\left(\boldsymbol{Q} \boldsymbol{W}_k^T \boldsymbol{E}_{l,:}^T \boldsymbol{M}\right) \boldsymbol{M}^T \boldsymbol{E}_{l,:} \boldsymbol{W}_v \\ &\Rightarrow \text{Softmax}\left(\boldsymbol{Q} \boldsymbol{K}'^T\right) \boldsymbol{V}' \\ &= \text{Softmax}\left(\boldsymbol{Q} \boldsymbol{W}_k'^T \boldsymbol{E}_{l,:}^T\right) \boldsymbol{E}_{l,:} \boldsymbol{W}_v'. \end{aligned} \tag{18}$$

Thus, we obtain (❸):

$$\begin{aligned} \boldsymbol{W}_k' &= \boldsymbol{E}_{l,:}^{\dagger} \boldsymbol{M}^T \boldsymbol{E}_{l,:} \boldsymbol{W}_k, \\ \boldsymbol{W}_v' &= \boldsymbol{E}_{l,:}^{\dagger} \boldsymbol{M}^T \boldsymbol{E}_{l,:} \boldsymbol{W}_v. \end{aligned} \tag{19}$$

This construction ensures the validity of Equation (11). In this context, $\boldsymbol{E}_{l,:}^{\dagger}$ indicates the *Moore–Penrose pseudoinverse* (Moore, 1920; Bjerhammar, 1951; Penrose, 1955) of $\boldsymbol{E}_{l,:}$.

Building upon the previous discussions (❶,❷,❸), we demonstrate the existence of a parameter set:

$$\{\boldsymbol{W}_q', \boldsymbol{W}_k', \boldsymbol{W}_v', \boldsymbol{W}_1', \boldsymbol{W}_2', b_1', b_2'\}$$

Table 4: Comparison of training techniques, where SFT, PO, and RL mean the abbreviation of supervised fine-tuning, preference optimization, and reinforcement learning, respectively.

| Techniques | Reasoning Trajectories | Outer Loop Loss |
|---|---|---|
| SFT | Off-policy | $\mathcal{L} = -\mathbb{E}_{(x,y)\sim\mathcal{D}}\left[\log p_\theta(y \mid x)\right]$ (Radford et al., 2018) |
| Off-Policy PO | Off-policy | $\mathcal{L} = -\log\sigma\left(r_\theta(y_{\text{preferred}}) - r_\theta(y_{\text{dispreferred}})\right)$ (Rafailov et al., 2023) |
| On-Policy RL | On-policy | $\mathcal{L} = -\mathbb{E}_t\left[\min\left(r_t(\theta)\hat{A}_t,\,\text{clip}(r_t(\theta), 1-\epsilon, 1+\epsilon)\hat{A}_t\right)\right]$ (Schulman et al., 2017) |

such that:

$$
\boldsymbol{W}_2^T\left(\sigma\left(\boldsymbol{W}_1^T\left(\text{Softmax}\left(\boldsymbol{E}_{t,0}\boldsymbol{W}_q\boldsymbol{W}_k^T\begin{bmatrix}\boldsymbol{E}_{l,:}\\\boldsymbol{E}_{t,0}\end{bmatrix}^T\right)\begin{bmatrix}\boldsymbol{E}_{l,:}\\\boldsymbol{E}_{t,0}\end{bmatrix}\boldsymbol{W}_v\right) + b_1\right)\right) + b_2 =
$$
$$
\boldsymbol{W}_2'^T\left(\sigma\left(\boldsymbol{W}_1'^T\left(\text{Softmax}\left(\boldsymbol{E}_{l,-1:}\boldsymbol{W}_q'\boldsymbol{W}_k'^T\boldsymbol{E}_{l,:}^T\right)\boldsymbol{E}_{l,:}\boldsymbol{W}_v'\right) + b_1'\right)\right) + b_2', \tag{20}
$$

which proves the Theorem 2.1. And this parameter set may not be the only viable option. For example, according to the universal approximation theorem (Cybenko, 1989; Hornik, 1991), a feed-forward network can be utilized to address differences in attention computations and provide a greater degree of freedom for $\boldsymbol{W}_q'$, $\boldsymbol{W}_k'$, and $\boldsymbol{W}_v'$.

$\square$

## E  INSTANTIATION OF TRAINING TECHNIQUES WITHIN RaML

We review various training techniques from a meta-learning perspective, including supervised fine-tuning (Howard & Ruder, 2018; Devlin et al., 2019), off-policy preference optimization (Rafailov et al., 2023), and on-policy reinforcement learning (Schulman et al., 2017; Ouyang et al., 2022; Shao et al., 2024; Ahmadian et al., 2024). We propose to categorize these techniques into two macro-level stages. The first stage involves acquiring reasoning trajectories and inputting them into the LLM to update its parameters $\theta$, thereby obtaining the output token distribution through updated $\theta$. Subsequently, the LLM parameters $\theta$ are optimized using a specific loss function based on this output distribution. Since various loss functions lead to the same maximum likelihood estimation (MLE) (Swamy et al., 2025), we attribute the essential difference between different training techniques to their inner loop optimization. Inner loop optimization is crucial in meta-learning training, as the meta-gradient is essential for enhancing meta-learning performance. Off-policy training techniques obtain reasoning trajectories through manual collection or synthesis, while on-policy training techniques generate reasoning trajectories based on the model distribution. From the perspective of learning to optimize, off-policy training techniques are equivalent to learning from an *optimal meta-optimizer*, directing the optimization of the inner loop. In contrast, RL requires independently exploring the inner loop's optimization path, presenting challenges due to increased freedom but allowing for potentially greater optimization outcomes.

## F  ADDITIONAL RELATED WORK

**LLM Reasoning.** The reasoning capabilities of large language models (LLMs) have progressively advanced through the development of several key technologies, which have substantially enhanced their performance on complex tasks. In-Context Learning (Brown et al., 2020; Rubin et al., 2022; Min et al., 2022b; Dong et al., 2024; Bertsch et al., 2024) enables models to perform tasks by interpreting examples provided in prompts without requiring additional training. However, this approach relies heavily on the model's pre-trained knowledge and careful prompt design, limiting its effectiveness for complex reasoning tasks (Min et al., 2022a). The introduction of Chain of Thought (CoT) (Wei et al., 2022; Yao et al., 2023; Besta et al., 2024) prompting has significantly improved LLM performance in areas such as mathematical reasoning, commonsense reasoning, and symbolic reasoning by guiding the models to produce intermediate reasoning steps. Supervised Fine-Tuning (SFT) further refines the reasoning capabilities of LLMs by training them with labeled datasets tailored to specific tasks (Beeching et al., 2024; Team, 2025b;a; Ye et al., 2025). Reinforcement Learning (RL), through the use of reward mechanisms, has become a critical approach to optimizing model behavior and enhancing reasoning abilities. Recently, Long-Chain of Thought (Long-CoT) models have emerged

as a notable trend in reasoning research, generating detailed reasoning steps to better address complex tasks (OpenAI, 2024b; DeepSeek-AI et al., 2025; Team et al., 2025; Team, 2025c; Deepmind, 2025).

**Additional Discussion on RAML and Related Work.** In this section, we aim to elucidate and discuss our work in comparison with several related studies. Dai et al. (Dai et al., 2023) interpret in-context learning (ICL) as LLMs generating meta-gradients from demonstration examples, which are applied to the base GPT model to construct an ICL system. In this work, each demonstration example serves as one data sample to update the parameters of LLMs. In contrast, our research emphasizes trajectory-aided reasoning, viewing each token as an update step and drawing extensive connections to supervised fine-tuning and reinforcement learning, rather than focusing on demonstration examples. Additionally, our approach incorporates more general training techniques with explicit parameter optimization, whereas ICL is constrained by limited demonstration examples, which poses certain limitations. Another research avenue explored by studies such as Gatmiry et al. (Gatmiry et al., 2024) and others (Huang et al., 2025) shows that transformers can learn to implement learning algorithms, such as gradient descent, within a chain of thought. However, these studies primarily investigate whether transformers can describe learning algorithms in natural language to solve practical numerical optimization problems, while our work delves into the internal parameter updates of the transformer model, applicable to a broader range of problems.

**Comparison with "Transformers Learn In-Context by Gradient Descent".** While our work shares the theoretical grounding with von Oswald et al. (2023) in viewing Transformer forward passes as gradient descent processes, we distinguish our approach by focusing on Chain-of-Thought reasoning rather than In-Context Learning. Whereas their framework relies on external context examples to drive the implicit optimization of a regression loss, RAML conceptualizes the generated reasoning tokens themselves as active pseudo-gradient updates that dynamically adapt the model parameters for specific queries. This distinction allows RAML to abstract the optimization concept beyond mechanistic linear regression to a broader meta-learning framework for complex reasoning, where the model functions as a meta-optimizer generating its own optimization path.

## G    IMPLEMENTATION DETAILS OF EXPERIMENTS

### G.1    IMPLEMENTATION DETAILS OF VISUALIZATION OF PSEUDO GRADIENT UPDATE

**Data Preparation.** We select four questions from AIME2024 as follows:

---

**Details of $q_0$**

QUESTION

Every morning Aya goes for a 9-kilometer-long walk and stops at a coffee shop afterwards. When she walks at a constant speed of $s$ kilometers per hour, the walk takes her 4 hours, including $t$ minutes spent in the coffee shop. When she walks $s + 2$ kilometers per hour, the walk takes her 2 hours and 24 minutes, including $t$ minutes spent in the coffee shop. Suppose Aya walks at $s + \frac{1}{2}$ kilometers per hour. Find the number of minutes the walk takes her, including the $t$ minutes spent in the coffee shop.

ANSWER

204

---

---

**Details of $q_1$**

QUESTION

There exist real numbers $x$ and $y$, both greater than 1, such that $\log_x\left(y^x\right) = \log_y\left(x^{4y}\right) = 10$. Find $xy$.

ANSWER

025

---

**Details of $q_2$**

QUESTION

Find the largest possible real part of

$$(75 + 117i)z + \frac{96 + 144i}{z}$$

where $z$ is a complex number with $|z| = 4$.

ANSWER

540

---

**Details of $q_3$**

QUESTION

Let $\triangle ABC$ have circumcenter $O$ and incenter $I$ with $\overline{IA} \perp \overline{OI}$, circumradius 13, and inradius 6. Find $AB \cdot AC$.

ANSWER

468

---

**Visualization of Pseudo Gradient Update.** We leverage QwQ-32B (Team, 2025c) to generate trajectories for these four questions. Then for each trajectory, we calculate the negative log-probability of `Final Answer\n\boxed..answer..` at each position. Algorithm 3 outlines the overall process.

---

**Algorithm 3:** Computation of Empirical Examples of Pseudo Gradient Update

**Input:** $\mathcal{M}$: QwQ-32B, $I$: instruction, $q$: question from AIME2024, $t$: the trajectory generated by QwQ-32B, $a$: the answer sequence, i.e., `Final Answer\n\boxed..answer..`, $s$: step size.

1 **for** $i \in [0, |t|)$ **do**
2    Obtain input by $I \oplus q \oplus t_{:i} \oplus a$ ;
3    Feed input to $M$ and get logits $l_a$ of answer sequence ;
4    Compute the negative log-probability using $l_a$ ;

---

**Visualization of Landscape.** We refer to the methodology proposed by Li et al. (Li et al., 2018). Assuming the set parameters of QwQ-32B is denoted by $\{\theta_k\}$ (excluding the embedding matrix), we randomly select two vectors, $\{\theta_{1,k}\}$ and $\{\theta_{2,k}\}$, for each parameter. We then edit the parameters by adding $\alpha_1\theta_{1,k} + \alpha_2\theta_{2,k}$ and compute the negative log-probability given only the instruction and question to form the point set $\{(\alpha_1, \alpha_2, \widehat{\mathcal{L}}_{\alpha_1,\alpha_2})\}$. Finally, we visualize this point set to reveal the landscape. The overall process is summarized as Algorithm 4.

---

**Algorithm 4:** Computation of Landscape

> **Input:** $\mathcal{M}$: QwQ-32B, $I$: instruction, $q$: question from AIME2024, $a$: the answer sequence, i.e., `Final Answer\n\boxed..answer...`

1 Obtain random vectors $\{\theta_{1,k}\}$, $\{\theta_{2,k}\}$ for each parameter $\theta_k$ of $\mathcal{M}$ ;
2 **for** $i \in [-1, 1, s]$ **do**
3      **for** $j \in [-1, 1, s]$ **do**
4          Get edited parameters $\{\theta'_k\}$ by adding $i \cdot \theta_{1,k} + j \cdot \theta_{2,k}$ ;
5          Obtain input by $I \oplus q \oplus a$ ;
6          Feed input to $M$ and get logits $l_a$ of answer sequence ;
7          Compute the negative log-probability using $l_a$ ;

---

**Project the Pseudo Gradient Update to Landscape.** To project the trajectory of the pseudo-gradient update onto the landscape, we fix one direction corresponding to the time dimension and identify the closest contour to the corresponding $\widehat{\mathcal{L}}$ to determine another direction.

### G.2 TRAINING DETAILS

**Dataset Processing.** To maintain the validity and verifiability of the question set, we clean and filter the original dataset. Initially, we exclude incomplete questions as well as those lacking answers. Subsequently, we remove questions requiring reasoning with images or other external information. To further ensure verifiability, we employ Math-Verify [2] to examine each question and exclude those that could not be verified. Finally, we eliminate irrelevant characters, such as URLs and HTML tags, resulting in approximately 39k questions with corresponding answers.

**Training of SFT.** We first synthesize training trajectories from Qwen2.5-Math-72B-Instruct (Yang et al., 2024b) and DeepSeek-R1-Distill-Qwen-14B (DeepSeek-AI et al., 2025). From the entire question set, we sample 10k questions and use the sampling parameters shown in Table 5 to generate reasoning trajectories with the prompt, *Please solve the following mathematical problem step by step and put your final answer in* $\backslash$*boxed*, resulting in 640k trajectories. We then filter out trajectories with

Table 5: Sampling parameters leveraging for reasoning trajectories synthesis.

| | **Qwen2.5-Math-72B-Instruct** | **DeepSeek-R1-Distill-Qwen-14B** |
|---|---|---|
| Temperature | 0.7 | 0.7 |
| Top-$p$ | 1.0 | 1.0 |
| Top-$k$ | 50 | 50 |
| Max Tokens | 8192 | 36784 |
| Rollout Number | 64 | 64 |

incorrect answers, retaining approximately $\sim 470$k for Qwen2.5-Math-72B-Instruct and approximately $\sim 550$k for DeepSeek-R1-Distill-Qwen-14B. During training, we utilize the parameters listed in Table 6.

---

[2] https://github.com/huggingface/Math-Verify

Table 6: Training parameters for SFT.

| | Parameter |
|---|---|
| Max Response Length | 18432 |
| Train Batch Size | 256 |
| Learning Rate | $1e$-5 |
| Total Epochs | 1 |

Table 7: Training parameters for GRPO.

| | Parameter |
|---|---|
| Max Prompt Length | 1024 |
| Max Response Length | 16384 |
| Rollout Temperature | 1.0 |
| Rollout Number | 16 |
| Train Batch Size | 1024 |
| Learning Rate | $1e$-6 |
| Total Epochs | 1 |

---

**System Prompt of GRPO**

A conversation between a User and an Assistant. The User poses a question, and the Assistant provides a solution. The Assistant's response follows these structured steps:

1. **Reasoning Process**: The Assistant reflects on the problem using a reasoning process enclosed within ¡think¿ and ¡/think¿ tags.
2. **Conclusion**: The Assistant reaches a conclusion, which is enclosed within ¡conclusion¿ and ¡/conclusion¿ tags. The final answer is highlighted within \boxed...final answer....
3. **Answer Format**: The complete response should be formatted as:
¡think¿
...reasoning process...
¡/think¿
¡conclusion¿
...conclusion...
The answer is \boxed...final answer...
¡/conclusion¿

---

**Training of GRPO.** For the GRPO training, we use the complete question set and apply the parameters listed in Table 7. We adhere to the DeepSeek-R1-style system prompt, as presented in the *System Prompt of GRPO* box. And for the reward design, we assign the trajectory with the correct answer and correct format the score 1, the trajectory with the false answer and correct format the score 0.0, trajectory with the correct answer and false format the score $-0.5$, and trajectory with the false answer and false format the score $-1$, formally:

$$R(y', y) = \begin{cases} 1 & \text{answer\_match}(y', y) \quad \text{and} \quad \text{format\_correct}(y'), \\ 0 & \neg\text{answer\_match}(y', y) \quad \text{and} \quad \text{format\_correct}(y'), \\ -0.5 & \text{answer\_match}(y', y) \quad \text{and} \quad \neg\text{format\_correct}(y'), \\ -1 & \neg\text{answer\_match}(y', y) \quad \text{and} \quad \neg\text{format\_correct}(y'), \end{cases} \tag{21}$$

where $y$ indicates the ground-truth and $y'$ indicates the trajectory. We employ the Math-Verify package to ascertain the equivalence of $y$ and $y'$.

**Details of Hardware and Software.** All the training tasks are conducted based on veRL (Sheng et al., 2025), cooperated with Pytorch (Paszke et al., 2019) 2.6.0, Transformers (Wolf et al., 2020) 4.51.3, vLLM (Kwon et al., 2023) 0.8.4. We conduct all experiments on clusters equipped with NVIDIA A800 GPUs and Intel(R) Xeon(R) Platinum 8336C CPUs.

### G.3 EVALUATION DETAILS

**Benchmarks.** The following details describe our evaluation benchmarks:

- **AIME24.** AIME24[3] consists of 30 challenging questions from the 2024 American Invitational Mathematics Examination (AIME).

---

[3] https://huggingface.co/datasets/AI-MO/aimo-validation-aime

- **MATH500.** The original MATH dataset (Hendrycks et al., 2021) comprises $12,500$ problems from American high school mathematics competitions. MATH500 (Lightman et al., 2024), a widely used subset of its test split, includes only Level 5 questions in this study.

- **LiveMathBench.** LiveMathBench (Liu et al., 2024) is a continuously updated dataset of challenging mathematical problems. We use the December 2024 hard split, comprising $45$ questions in English and Chinese.

- **GPQA.** GPQA (Rein et al., 2023) dataset is a challenging, professional multiple-choice science question-answering dataset. We use its diamond subset, comprising $198$ questions.

- **LiveCodeBench.** LiveCodeBench (Jain et al., 2024) is a benchmark designed for a comprehensive and uncontaminated evaluation of the code-related capabilities of LLMs. It incorporates questions from LeetCode, AtCoder, and Codeforces.

**Metrics.** We use Pass@$k$ and mG-Pass@$k$ (Liu et al., 2024) as evaluation metrics. We generate $n$ responses for each question and assume the number of correct responses is $c$. Then the metrics are computed as:

- **Pass@$k$.**

$$\text{Pass@}k = \mathbb{E}_{\text{questions}} \left[ 1 - \frac{\binom{n-c}{k}}{\binom{n}{k}} \right]. \tag{22}$$

- **mG-Pass@$k$.**

$$\text{mG-Pass@}k = \mathbb{E}_{\text{questions}} \left[ \frac{2}{k} \sum_{i=\lceil k/2 \rceil + 1}^{k} \sum_{j=i}^{c} \frac{\binom{c}{j} \cdot \binom{n-c}{k-j}}{\binom{n}{k}} \right]. \tag{23}$$

### G.4 MODELS UTILIZED IN FIGURE 3

The evaluation includes prominent models such as Sky-T1-32B (Team, 2025a), Bespoke-Stratos-32B (Labs, 2025), LIMO (Ye et al., 2025), s1.1-32B (Muennighoff et al., 2025), OpenThinker-32B (Team, 2025b), Light-R1-32B (Wen et al., 2025), DeepSeek-R1-Distill-Qwen-32B (DeepSeek-AI et al., 2025), DAPO-32B (Yu et al., 2025), and VAPO-32B (Yue et al., 2025). These models are based on either Qwen2.5-32B or Qwen2.5-32B-Instruct only through SFT or RL (**Zero-RL**). Since VAPO is not open source, we copy its results from the original paper.

## H MORE DISCUSSIONS ON RECENT LLM REASONING PROGRESS

In this section, we focus on recent research developments and discuss the essential improvements they implemented to enhance performance within our framework. We involve the following representative works: OpenThoughts (Team, 2025b), Light-R1 (Wen et al., 2025), Open-Reason-Zero (Hu et al., 2025), DAPO (Yu et al., 2025), VAPO(Yue et al., 2025), GPG (Chu et al., 2025b), Llama Nemotron (Bercovich et al., 2025).

**Data Filtering.** Works such as Light-R1 (Wen et al., 2025) use strategies like diversity and difficulty filtering to obtain high-quality data. From a meta-learning perspective, this approach can be seen as a sample mining strategy, optimizing the distribution of training task sets to enhance the efficiency of model training.

**Synthetic Data From Strong Reasoning LLMs.** Works such as OpenThoughts (Team, 2025b) and Llama Nemotron (Bercovich et al., 2025) utilize a more advanced reasoning LLM, such as DeepSeek-R1, to generate multiple trajectories for each training question, resulting in training data for SFT. This approach effectively expands the size of the support set to stabilize inner loop optimization, thereby achieving improved results. On the other hand, this is equivalent to distilling the optimization path from the already trained model (strong reasoning LLMs) to the small model.

**Clip Higher for Clipper Surrogate Loss of RL.** DAPO (Yu et al., 2025) proposes using a higher clipping range to promote exploration during the GRPO training process. Similarly, removing the KL penalty term and entropy loss in GRPO can achieve the same effect. From an optimization

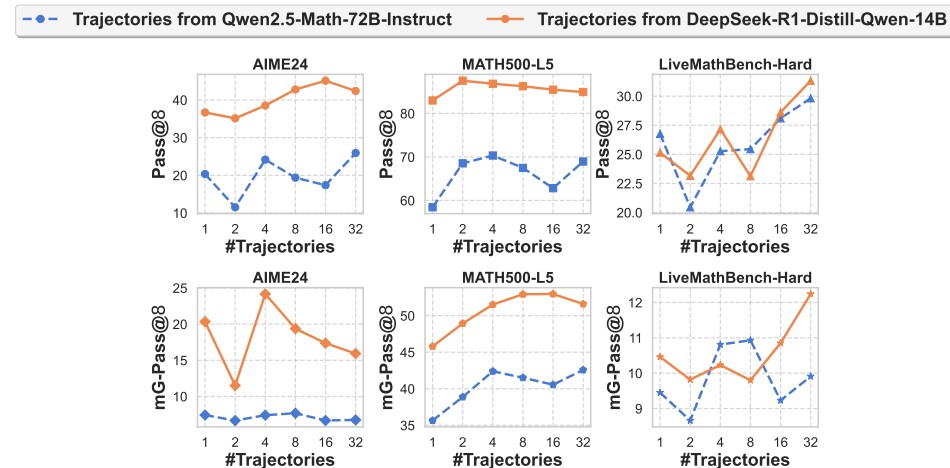

Figure 9: Evaluation results *w.r.t.* different number of reasoning trajectories of SFT models on AIME24, MATH500-L5, and LiveMathBench-Hard.

perspective, these improvements expand the exploration space of the optimization path, facilitating the model's ability to explore extreme points. Increasing the diversity of training data in supervised fine-tuning also contributes to this effect.

**Dynamic Sampling During RL.** Recent studies (Chu et al., 2025b; Yu et al., 2025) aim to balance the ratio of correct to incorrect trajectories during rollout by employing dynamic sampling or introducing bias. This strategy equalizes the positive and negative gradients in the inner loop, thereby alleviating model overfitting to a particular class.

**Group-Sampling for PPO.** In classic reinforcement learning methodologies, algorithms typically generate only a single trajectory per problem instance. Recent advancements (Yue et al., 2025; Hu et al., 2025) have introduced group sampling in algorithms such as PPO, allowing the generation of multiple trajectories for each problem. From the perspective of this study, this improvement corresponds to expanding the support set, thereby enhancing inner-loop optimization.

## I  ADDITIONAL EXPERIMENTS

### I.1  MANIPULATING TRAINING REASONING TRAJECTORIES PER QUESTION TO ENHANCE LLM REASONING

Previous studies (Agarwal et al., 2021; Chen et al., 2020a) highlight that the size of the support set is of paramount importance in improving performance, stability, and convergence in meta-learning. In RAML, the support set is intrinsically connected with the number of reasoning trajectories trained per question, prompting the question: *Can enhancements in support set size contribute to more effective training in LLM reasoning?* In this preliminary study, we investigate the impact of increasing the number of training reasoning trajectories per question on LLM reasoning.

**SFT.** We train Qwen2.5-7B-Base through SFT with $\{1, 2, 4, 8, 16, 32\}$ synthetic reasoning trajectories, ensuring equal training frequency per question. Evaluation results (Figure 9) show that increasing the number of trajectories improves performance and reasoning stability across all benchmarks, suggesting that additional trajectories enhance supervised fine-tuning outcomes (Yang et al., 2024b).

**GRPO.** With regards to GRPO, the support set size corresponds to the number of trajectories in the rollout group for each prompt (question). To maintain stable advantage estimation in GRPO and ensure a fair comparison, we fix the rollout group size at 16 and calculated the advantage for each trajectory. During gradient updates, however, we randomly select $n \in \{1, 2, 4, 8, 16\}$ trajectories to calculate the gradient. Experimental results shown in Figure 10 demonstrate that: 1) multiple trajectories for a single question significantly enhance model performance and stability; 2) a larger

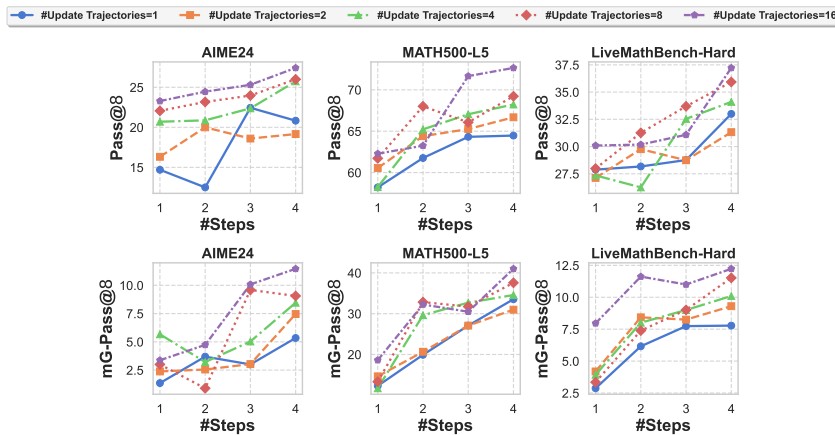

Figure 10: Evaluation results *w.r.t.* different number of reasoning trajectories of GRPO models on AIME24, MATH500-L5, and LiveMathBench-Hard.

number of trajectories accelerates convergence. These findings explain the superior performance and stability of GRPO-based (or other similar RL algorithms) reasoning models (Liu et al., 2024), as the GRPO mechanism inherently optimizes for multiple trajectories per question compared with PPO (Schulman et al., 2017) and naive SFT. It is noteworthy that some studies have attempted to enhance PPO through group sampling (Hu et al., 2025; Yue et al., 2025) and achieve competitive performance compared with original PPO.

### I.2 INCENTIVIZING REASONING EFFICIENCY BY OPTIMIZATION LENS

Recent advanced reasoning models face limitations due to inefficient and excessively lengthy reasoning trajectories. Although several studies (Ma et al., 2025; Yang et al., 2025a) have attempted to minimize the number of decoding tokens to mitigate overhead, these approaches frequently lead to decreased reasoning performance, thereby presenting a fundamental question: *Can we reduce the number of reasoning tokens without compromising reasoning performance?* As previously discussed, each reasoning trajectory corresponds to an inner-loop optimization trajectory, thus reframing the inquiry as follows: *Can there be a s more effective inner loop optimization path?* From an optimization perspective, as illustrated in Figure 11, there exists such an optimal inner loop optimization path. In this section, we present a straightforward yet convincing experiment to validate the existence of this inner loop optimization path.

Specifically, we employ Qwen3-32B (Yang et al., 2025a) to generate 16 reasoning trajectories for each question in AIME24, MATH500-L5, and LiveMathBench-Hard. These trajectories serve as foundational optimization paths, and our goal is to refine them to discover more optimal paths. We propose an heuristic method which condense the reasoning trajectories by using an LLM to summarize the original trajectories into shorter variants, which are then used to prompt Qwen3-32B for answer generation. The summarizations are generated by Qwen2.5-32B-Instruct, deliberately excluding answers from the summarized reasoning trajectories. We performed four summary generators to reduce the impact of randomness. Figure 12 displays the experimental results. Notably, we observe that Qwen3-32B's performance with summarized reasoning trajectories is comparable to its performance in thinking mode especially for the Pass@16 metric, while significantly reducing the number of tokens in the reasoning trajectories. Moreover, Qwen3-32B's performance using summarized reasoning trajectories surpasses that in no-thinking mode, even that the latter has more tokens.

Our experiments demonstrate that trained long-CoT LLMs have the potential to achieve optimal reasoning trajectories that require fewer tokens while maintaining comparable reasoning performance. We approximate these trajectories using a straightforward method, leaving the exploration of more advanced approaches for future work.

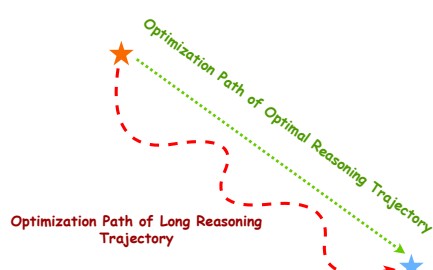

Figure 11: Given a long reasoning trajectory, there exists an optimal corresponding reasoning trajectory which leverage less tokens.

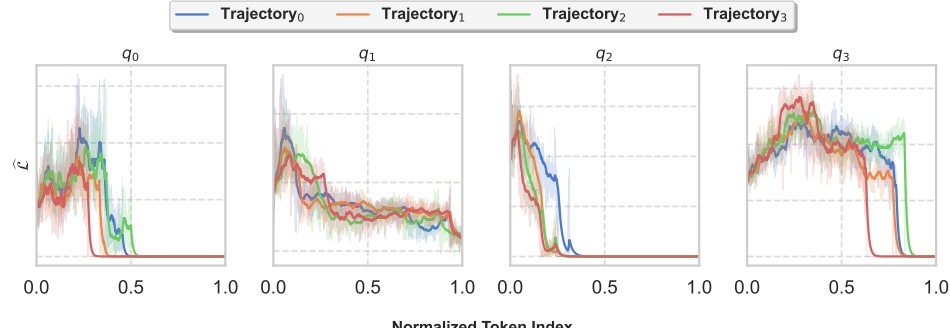

Figure 12: Experimental results of Qwen3-32B with 1) thinking mode, 2) summarized trajectory, and 3) nothinking mode on AIME24, MATH500-L5, and LiveMathBench-Hard-en.

Figure 13: Visualization of the *pseudo-gradient update*: The $x$-axis represents the normalized indices of corresponding trajectories. $q_0, q_1, q_2, q_3$ are question selected from AIME24, refer to § G.1 for more details.

### I.3 DETAILS OF PSEUDO GRADIENT UPDATE

**QwQ's Pseudo Gradient Update.** Following the methodology described in § 2.2 and § G.1, we monitor the pseudo-gradient update of QwQ-32B. As shown in Figure 13, the negative log-probability progressively decreases along the reasoning trajectories which aligns with our definition.

**Qwen3's Pseudo Gradient Update.** Following the methodology described in § 2.2 and § G.1, we monitor the pseudo-gradient update of Qwen3-32B (Yang et al., 2025a) under thinking mode, as illustrated in Figure 14. We observe that the reasoning trajectories of Qwen3 exhibit a parameter update effect.

**Qwen3's Pseudo Gradient Update in Thinking/NoThinking Mode.** Referring to the settings in § 3.3, we examine the differences between thinking mode and no-thinking mode, as shown in Figure 15. It is evident that, due to the specific optimization of Qwen3, its no-thinking token delimiter (i.e., `</think>`) demonstrates a more pronounced gradient descent effect. The delimiter `</think>` enables the model to swiftly update to an extreme point in the appropriate direction with a larger step size. However, this update is susceptible to falling into local minima, which accounts for the performance gap between Qwen3's no-thinking mode and thinking mode.

**Pseudo Gradient Update of False Reasoning Trajectories.** Figure 16 illustrates the curve of pseudo gradient updates associated with incorrect reasoning trajectories. It is evident that the curve representing these trajectories does not show a downward trend, underscoring the strong connection between reasoning paths and optimization processes.

## J FUTURE DIRECTIONS

Further studies can be conducted based on our work, for instance:

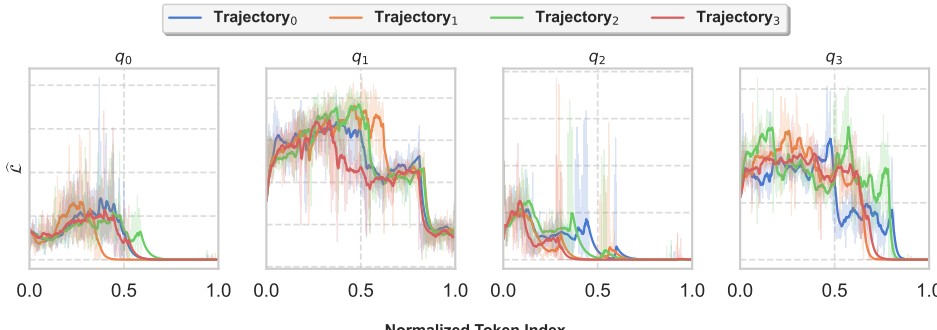

Figure 14: Visualization of the *pseudo-gradient update*: The $x$-axis represents the normalized indices of corresponding trajectories. $q_0, q_1, q_2, q_3$ are question selected from AIME24.

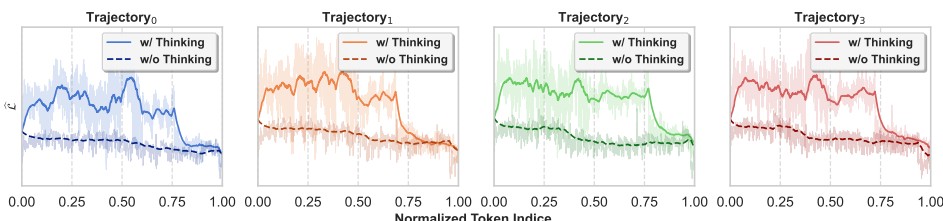

Figure 15: Illustration of Qwen3's *pseudo-gradient update* for both thinking and non-thinking modes. We visualize four pairs of correct reasoning trajectories for one question in AIME24.

- **Further Understanding Reasoning Trajectories**

  ❶ Unlike typical meta-learning frameworks with predefined support sets, the reasoning trajectories in LLMs are self-generated. This implies that LLMs inherently learn gradient update steps without needing explicit support sets for fine-tuning. *Investigating how LLMs learn to form effective reasoning trajectories, namely, gradient update steps*, presents an intriguing challenge.

  ❷ Tokens contribute differently to the modification of model parameters. *What accounts for this disparity among tokens? Is it connected to their semantic properties, and if so, in what manner*?

  ❸ Trajectory-aided reasoning in large language models (LLMs) demonstrates comparable generalization abilities across various tasks. *What aspects of the learning process contribute to this generalization ability, and which meta-features are developed through the optimization of reasoning trajectories?*

- **Towards Enhancing LLM Reasoning**

  ❶ *Improved Reasoning Trajectory Selection Strategy in LLM Training.* During both supervised fine-tuning and reinforcement learning, reasoning trajectories usually remain constant. Could implementing an adaptive sampling mechanism, similar to those utilized in meta-learning (Yao et al., 2021a; Liu et al., 2020), enhance training efficacy?

  ❷ *Enhancing Reasoning Efficiency Through an Optimization Perspective.* Given that each token offers a unique contribution to optimization, is there a strategy to discern these contributions to filter out superfluous tokens, thereby improving reasoning efficiency?

  ❸ *Task (Question) Ratio to Enhance Generalization Across Different Domains.* Insights from related studies, such as Collins et al. (Collins et al., 2020) and Wang et al. (Wang et al., 2022), in meta-learning, suggest methods to bolster the reasoning capability of LLMs, enabling them to generalize across domains—for instance, training on mathematical data and inferring insights from coding data.

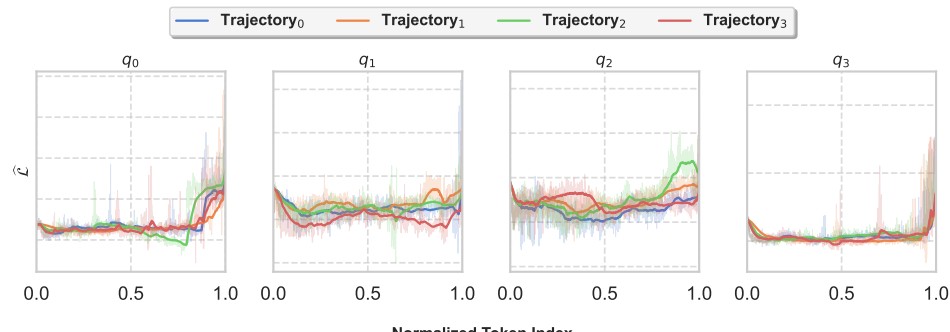

Figure 16: Illustration Qwen3's *pseudo-gradient update* corresponding to false reasoning trajectories.

## K    LIMITATIONS

**Limitations of Theoretical Analysis.**  This paper's theoretical analysis, while providing a foundational framework, relies on several approximate assumptions, as discussed in Equation (17). Although these assumptions hold valid in most typical scenarios, a more detailed investigation into their specific conditions and potential limitations is warranted in subsequent research to fully understand their implications. Additionally, while the theory presented herein confirms the existence of the related equation cited in Theorem 2.1, deriving its precise analytical solution remains an open problem requiring further exploration. Finding this solution could potentially lead to a deeper theoretical understanding of the pseudo-gradient dynamics.

**Limitations of Experiments.**    Firstly, due to time and resource constraints, our experiments primarily focused on mathematical reasoning tasks. While mathematical reasoning shares many commonalities with other reasoning tasks, suggesting our conclusions may generalize, differences in the specific optimization behavior across diverse task types warrant further dedicated study. Secondly, experiments were conducted on a limited selection of LLMs. Although the observed performance behavior across these models was largely consistent, indicating the general applicability of our conclusions, future work should investigate a wider range of LLM architectures to identify any subtle, model-specific differences. Finally, mirroring a limitation discussed previously, our experiments were confined to the textual modality. Given that recent research indicates the significance of reasoning trajectories in multimodal contexts, exploring their role through experiments involving multimodal data is a valuable direction.

## L    LLM USAGE

In this study, the deployment of LLMs is intentionally limited to the final phases of our research, specifically for refining and proofreading the manuscript. LLMs are utilized solely to enhance the clarity, logical coherence, and linguistic precision of the narrative, ensuring a clear and sophisticated articulation of our ideas. Crucially, LLMs played no role in the foundational aspects of this investigation, including the formulation of the research strategy, the design of the experimental framework, or the interpretation of the results.

