# OpenReview forum: "Reasoning as Meta-Learning: An Optimization Perspective to Decipher Long CoT Reasoning in LLMs"
_ICLR.cc/2026/Conference — Submitted to ICLR 2026_

### Official Review · Reviewer_zmaQ · 2025-10-30

**Soundness:** 1
**Presentation:** 1
**Contribution:** 1
**Rating:** 2
**Confidence:** 3

**Summary:**

The paper proposes Reasoning as Meta-Learning as a theoretical and empirical framework that interprets

LLM reasoning as a meta-learning process. It conceptualizes a model’s reasoning trajectory (chain-of-thought) as a pseudo-gradient descent path that updates the model’s hidden parameters while reasoning, analogous to how inner-loop optimization operates in Model-Agnostic Meta-Learning (MAML). The authors then utilize the conceptual framework to explain LLM reasoning.

**Strengths:**

The work proposes a novel conceptual framework. The association between meta-learning and ICL is interesting.

**Weaknesses:**

The conceptual framework is quite disconnected to experiments. The experimental results are mostly past works’ finding, while the authors explain them in their own words using the proposed framework. However, it is poorly explained. Many places are very unclear, especially inner-loop, outer-loop in MAML and how is the optimization working here for LLMs.

 The framework is over-simplified, while actual LLMs are more complex. No sufficient discussion or theory to fill this gap in this paper.

The authors assume longer CoTs lead to better reasoning results and the model optimizes its result after each step (the authors assume the answer is always improving). However, this assumption is too strong, not well established empirically or no theoretical guarantee.  Many past works are showing the opposite. The models can give correct answers without CoT or in the middle of CoT. The models may even give right early-exit answers despite wrong reasoning steps.

for example, Lanham, Tamera, et al. "Measuring faithfulness in chain-of-thought reasoning." *arXiv preprint arXiv:2307.13702* (2023). Yang, Chenxu, et al. "Dynamic Early Exit in Reasoning Models." *arXiv preprint arXiv:2504.15895* (2025)…

Notations are not very clearly defined.


**Suggested references to add:**

Transformers Learn In-Context by Gradient Descent

This work also introduce the idea of pseudo gradient. the reasoning steps in CoT can be viewed as context as well before the answer. The authors should discuss this work in literature review.

**Questions:**

what is W1, W2? There is also no formal definition of L

---

> ### Author Response · Authors · 2025-11-21
> **Response to Reviewer zmaQ**
>
> Dear reviewer zmaQ,
>
> Thank you for your detailed review. Below, we will address your concerns.
>
> > Response to **W1: Framework Disconnection**
>
> We acknowledge that LLMs are complex systems. However, the value of RAML lies in providing a high-level understanding that unifies various phenomena (SFT stability vs. RL exploration, CoT length) under a unified explanative umbrella. The contributions of this paper are primarily **empirical validation** of this perspective (Section 3) and **practical application** of meta-learning concepts to improve reasoning (Section 4), rather than detailed mechanistic derivation.
>
> > Response to **W2: Assumption on Longer CoTs**
>
> We do **not** assume that the answer probability improves monotonically at every step. As shown in **Figure 4** and **Figure 13**, confidence fluctuates during the reasoning process. Our claim is that longer trajectories provide *more optimization steps*, thereby increasing the probability of escaping local minima (hallucinations) and converging to a correct solution. This is consistent with optimization theory where more steps allow for better convergence, even if the loss surface is non-convex.
>
> > Response to **W3: Missing Literature**
>
> Thank you for pointing out "Transformers Learn In-Context by Gradient Descent." We have discussed related ICL works in **Appendix F**, but we will explicitly add a detailed discussion of this specific paper to contextualize our focus on *Reasoning Trajectories* versus *Contextual Examples* in the revised version.
>
> > Response to **Q1: Definitions**
>
> - **$W_1, W_2$:** These represent the weight parameters of the Feed-Forward Network (FFN) within the Transformer layer, as defined in **Equation (5)** and **Appendix A**.
> - **$\mathcal{L}$ (Objective):** $\mathcal{L}$ represents the abstract objective function of the reasoning task. Since the true objective is intractable in discrete generation, we use the negative log-probability of the correct answer as a proxy observable, a standard approach in interpretability research.
> ---
>
> Thanks again for your review. We hope our reply can address your questions and concerns, and we are also open to additional questions.

---

### Official Review · Reviewer_3CaD · 2025-11-01

**Soundness:** 2
**Presentation:** 1
**Contribution:** 2
**Rating:** 2
**Confidence:** 4

**Summary:**

This paper proposes RAML (Reasoning As Meta-Learning), a novel framework for interpreting LLM reasoning capabilities through meta-learning principles. The key insight is conceptualizing reasoning trajectories (chain-of-thought steps) as pseudo-gradient descent updates to the LLM's parameters. The authors formalize reasoning task training as meta-learning, where each question is a task, reasoning trajectories serve as inner-loop optimization for parameter adaptation, and answers constitute the query set. The paper provides theoretical analysis (proving existence of parameter updates corresponding to trajectory tokens), extensive empirical validation on mathematical reasoning benchmarks, and practical insights for improving LLM reasoning through meta-learning techniques.

**Strengths:**

1. The mapping between training approaches (SFT vs. RL) and meta-learning concepts (off-policy vs. on-policy optimization, support set size) provides potentially new theoretical insights into recent advances in long-CoT reasoning models.
2. The experimental design is sound—training models from scratch to avoid confounds from prior training. Multiple benchmarks are employed, and the visualization of loss landscapes and pseudo-gradient updates provides compelling, intuitive evidence. The ablation studies systematically examine key factors, and practical methods are proposed and validated.
3.  The framework opens connections to the rich meta-learning literature, suggesting numerous research directions. The finding about optimal shorter trajectories maintaining performance has important practical implications for inference efficiency.

**Weaknesses:**

1. The paper analyzes reasoning models through a meta-learning lens. However, it lacks a clear articulation of how this framework fundamentally differs from existing mechanistic analyses of transformers' learning capabilities via attention mechanisms. Specifically, the relationship to prior work on mesa-optimization [1], implicit dynamics of in-context learning [2], and higher-order optimization in ICL [3] remains unclear. The paper must establish stronger motivation for why this particular analytical framework is essential and demonstrate concrete advantages over existing frameworks for understanding transformers' implicit learning properties.
2. Several critical aspects of the proposed framework are underspecified or unsound:
- The paper borrows terminology from meta-learning and optimization theory without establishing rigorous connections. Specifically, how does a classical learned optimizer (e.g., meta-Adam, meta-SGD) formally correspond to a reasoning model that generates reasoning trajectories? This equivalence requires explicit justification.
- Ambiguities in Algorithm 2  (Appendix C): For each question i, multiple trajectories t_j are generated. However, the granularity is unclear. Does a trajectory represent a single token or a token sequence? Section 3.3 states that each token corresponds to one optimization step, which would imply N×T inner-loop optimization steps for T trajectories of N tokens each. Additionally, is optimization performed in a mini-batch or a sequential format? These details are crucial for reproducibility.
- Since the pseudo-gradient update lacks explicit evaluation and backward passes, and no explicit parameter updates occur, the second-order gradient must be approximated rather than computed via automatic differentiation. The approximation method and its implications require explicit discussion.
3. The paper lacks an essential theoretical analysis of the proposed meta-learning framework. Specifically, there is no proof of convergence for the inner-loop optimization over reasoning trajectories. There is no analysis of generalization or smoothness properties. The pseudo-gradient approximation of second-order gradients (versus full backpropagation) introduces bias and computational differences that demand rigorous characterization.
4. The framework exclusively relies on pseudo inner-loop objectives based on model log-likelihood. This raises concerns about error propagation when initial model likelihoods are inaccurate. The framework should consider a broader range of objectives and optimization methods within the meta-learning paradigm, such as energy function minimization [4,5] and meta-reinforcement learning approaches like Meta-CoT [6].
5. Several key takeaways, e.g., "SFT leads to stable reasoning rollouts" and "Longer reasoning trajectories improve performance", have been extensively validated in prior work. While confirmation may have some value, these findings provide minimal marginal contribution to the field.

### References
[1] Von Oswald, J., Schlegel, M., Meulemans, A., Kobayashi, S., Niklasson, E., Zucchet, N., ... & Sacramento, J. (2023). Uncovering mesa-optimization algorithms in transformers. arXiv preprint arXiv:2309.05858.
[2] Dherin, B., Munn, M., Mazzawi, H., Wunder, M., & Gonzalvo, J. (2025). Learning without training: The implicit dynamics of in-context learning. arXiv preprint arXiv:2507.16003.
[3] Fu, D., Chen, T. Q., Jia, R., & Sharan, V. (2024). Transformers learn to achieve second-order convergence rates for in-context linear regression. Advances in Neural Information Processing Systems, 37, 98675-98716.
[4] Gladstone, A., Nanduru, G., Islam, M. M., Han, P., Ha, H., Chadha, A., ... & Iqbal, T. (2025). Energy-Based Transformers are Scalable Learners and Thinkers. arXiv preprint arXiv:2507.02092.
[5] Oarga, A., & Du, Y. (2025). Generalizable Reasoning through Compositional Energy Minimization. arXiv preprint arXiv:2510.20607.
[6] Xiang, V., Snell, C., Gandhi, K., Albalak, A., Singh, A., Blagden, C., ... & Finn, C. (2025). Towards system 2 reasoning in llms: Learning how to think with meta chain-of-thought. arXiv preprint arXiv:2501.04682.

**Questions:**

See my questions in Weaknesses.

---

> ### Author Response · Authors · 2025-11-21
> **Response to Reviewer 3CaD**
>
> Dear review 3CaD,
>
> Thank you for recognizing that our paper has sound experimental design and insightful numerous research directions. We respond to your concerns below.
>
> > Response to **W1: Distinction from ICL & Mechanism Analysis**
>
> We discuss the relationship with ICL literature in **Appendix F**:
>
> - **ICL Research (e.g., Dai et al.):** Views demonstration examples in the prompt as data points for meta-gradient updates.
> - **RAML (Our Work):** Focuses on **Chain-of-Thought** reasoning. We conceptualize the *self-generated reasoning tokens* themselves as the update steps. This allows us to analyze modern reasoning training paradigms (SFT and RL) where no demonstration examples are provided, which prior ICL theories do not cover.
>
> > Response to **W2.1: Issues About Establishing Rigorous Connections Between ML and LLM Reasoning**
>
> We established the connection between the generation of CoT and parameter updates in Proposition 2.1. We agree that the subsequent connection with meta-learning is largely based on intuition, but the main purpose of this paper is not to derive the relationship between the two through detailed theoretical reasoning, because such derivation is non-trivial due to the large number of parameters and complex internal mechanisms of LLMs. The main purposes of this paper are 1) **to verify the connection between meta-learning and CoT reasoning through extensive experiments (Section 3)**; 2) further, **we hope to verify that existing achievements and research in meta-learning can successfully guide the development of LLM reasoning (Section 4 & Appendix)**.
>
> > Response to **W2.2: Algorithm 2 Ambiguities:**
>
> In **Algorithm 2**, a "trajectory" $t_j$ refers to a complete sequence of reasoning tokens. During the inner loop, the model generates this sequence. Each token generation conceptually represents an optimization step. The outer loop optimization (training) is performed in mini-batches over these trajectories, updating the model parameters $\theta$ to maximize the likelihood of the correct answer given the trajectory.
>
> > Response to **W2.3 & W3: Theoretical Convergence & Second-Order Gradients**
>
> RAML is a **descriptive framework** to interpret LLM behaviors, not a prescriptive mathematical proof of convergence. We aim to show the *feasibility* of the pseudo-gradient view to explain empirical phenomena (e.g., why longer CoT works). Explicitly computing second-order gradients is computationally prohibitive and unnecessary for the analogy to hold; standard backpropagation in SFT/RL serves as the implementation of the outer-loop update, implicitly optimizing the "optimizer" (the LLM).
>
> > Response to **W4: Proxy Objectives**
>
> Using log-likelihood of the answer as a proxy for the objective is standard practice in interpreting language models (e.g., probing confidence) [1][2]. Since reasoning is a discrete generation process without a differentiable "true" objective, log-likelihood is the most principled accessible metric.
>
> > Response to **W5: Novelty of Takeaways**
>
> Our takeaways do not point out some existing conclusions, but rather explain and understand the experimentally observed conclusions in the context of meta-learning. While some individual findings (e.g., "longer CoT is better") exist in isolation, RAML provides a **unified explanation** for *why* they occur (longer CoT = more inner loop optimization steps).
>
> Furthermore, as noted in our response to Reviewer dnUv, the insight regarding **Support Set Size** (training with multiple trajectories per question) is a direct, novel derivative of our meta-learning mapping that has not been a standard focus in SFT literature.
>
> [1] Zhou, Xiangxin, et al. "Reinforcing general reasoning without verifiers, *arXiv preprint* 2025.
>
> [2] Yu, Tianyu, et al. "RLPR: Extrapolating RLVR to General Domains without Verifiers." *arXiv preprint 2025*.
>
> ---
>
> Thank you for your diligent review. We hope our reply can address your questions and concerns, and we are also open to potential other questions.

---

### Official Review · Reviewer_PWxJ · 2025-11-01

**Soundness:** 3
**Presentation:** 3
**Contribution:** 3
**Rating:** 6
**Confidence:** 3

**Summary:**

The authors in thia paper reinterpret long chain-of-thought (CoT) reasoning in large language models (LLMs) as a form of meta-learning. Specifically, they treat reasoning trajectories as optimization: Each reasoning step is treated as an implicit parameter update, akin to gradient descent, guiding the model toward the correct answer. And provide both theoretical analysis and empirical results for this claim. Then, they conceptualize that each question is framed as a separate task. The reasoning trajectory acts as the inner-loop optimization during meta learning, and the final answer is the outer-loop objective.

They observed the following findings:
- Supervised fine-tuning (SFT) provides stable inner-loop optimization, while reinforcement learning (RL) offers exploration benefits. Combining both yields better results.
- Longer reasoning trajectories improve performance, similar to more optimization steps in meta-learning.
- Reflection tokens (e.g., “Wait”, “Therefore”) act like optimization triggers, helping models escape local minima.
- Generalization: Models trained with trajectories generalize better across domains (math, science, code), showing transferability of reasoning skills.

**Strengths:**

The paper introduces a meta-learning perspective on chain-of-thought reasoning, viewing each reasoning trajectory as a pseudo-gradient descent inner-loop update. This conceptual reframing is novel compared to [other work](https://aclanthology.org/2023.findings-acl.247.pdf) that view in-context learning as meta learning.

The paper is easy to follow. Authors provide takeaways and discuss implications to reasoning training in section 4.

**Weaknesses:**

Training uses Open Reasoner Zero dataset with synthetic trajectories from large teachers; evaluation is said to be orthogonal to training data. Did the author check for contamination between ORZ/teachers and AIME24, MATH500, LiveMathBench, GPQA, LiveCodeBench?

Authors present this paper with core theoretical premise: each reasoning token induces a pseudo-gradient update to model parameters/ However such a premise is only partially formalized. While Proposition 2.1 provides a mapping, the paper is short of: analyzing approximation error, quantifying the degree of equivalence, specifying conditions under which the pseudo-update interpretation holds.

Missing error bars/CI tests across main tables/figures. Please add appropriate information about statistical significance (error bars, confidence intervals, statistical significance tests) and details about how this was computed.

**Questions:**

Most experiments use math-focused datasets (AIME, MATH500, LiveMathBench). While the authors test GPQA and code reasoning, the coverage is shallow. This narrows the generality claim, especially since reasoning difficulty distribution differs across domains. How does the method apply to reasoning in other domains, such as [medical](https://aclanthology.org/2025.acl-long.896/) and [law](https://arb.duckai.org/).

The pseudo-gradient interpretation would be more convincing if supported by probing internal activations. Would be great if authors could include representational similarity analysis or [causal tracing](https://www.anthropic.com/research/tracing-thoughts-language-model).

Can gradient conflict resolution in meta learning inspire weighted trajectory blending in post training LLM? Meta-learning has the gradient interference problem. Conflicting tasks lead to brittle meta-initialization, with algorithms like PCGrad, GradVac, Fishr address this. Can this leads to some hidden implication that reasoning trajectories include conflict: deductive vs heuristic, long reflection vs direct derivation.

Authors mentioned that special ENDOFTHINKING tokens regulating the length of reasoning facilitate fast-converging optimization steps. But when should we insert ENDOFTHINKING tokens (early stop of reasoning) in the reasoning trajectory to make efficient reason: early stops can reach the same answer as continue thinking?

---

> ### Author Response · Authors · 2025-11-21
> **Response to Reviewer PWxJ**
>
> Dear reviewer PWxJ,
>
> Thank you for finding our work innovative and accessible. We appreciate your constructive feedback and have addressed your concerns below.
>
> > Response to **W1: Data Contamination**
>
> We rigorously verified no overlap between the training data (ORZ, filtered to 39k questions as detailed in Section G.2) and evaluation benchmarks. ORZ is a well-known and community-standard dataset designed to minimize leakage, and our checks confirm orthogonality.
>
> > Response to **W2: Theoretical Rigor of Pseudo-Gradient Update**
>
> Our primary goal was to establish a conceptual bridge between forward computation and pseudo-gradient updates to motivate the RAML framework. **Proposition 2.1** demonstrates the theoretical feasibility of this mapping within a standard Transformer layer. Given that modern LLMs are stacks of such layers, this proposition holds structurally. We acknowledge that quantifying the exact approximation error is non-trivial due to the abstract nature of the implicit objective function in natural language generation and complex internal structure of LLMs, but the empirical alignment (**Figures 2 and 13**) supports the validity of this perspective.
>
> > Response to **W3: Error Bars**
>
> We agree that statistical significance is important. In all reported experiments, we conducted multiple runs and reported average performance to mitigate variance (e.g., Pass@8 from 8 samples per question, as in Section G.3). We will include explicit error bars and confidence intervals in the revised revision to further strengthen the results.
>
> > Response to **Q1: Domain Generality**
>
> We acknowledge the value of broadening the range of benchmarks. Our evaluation currently covers Mathematics, STEM (GPQA), and Coding. We believe these domains effectively represent complex reasoning tasks. Notably, GPQA already encompasses diverse scientific disciplines including biology, chemistry, and physics. More importantly, these three datasets are benchmarks that are widely used and recognized in research work related to LLM reasoning, and the capabilities examined by benchmarks in other vertical fields are all covered by these benchmarks.
>
> > Response to **Q2: Probing Internal Activations**
>
> While causal tracing is a valuable technique, it requires access to internal model states, which restricts analysis to open-source models. We deliberately chose **logits** (negative log-probability) as our primary probe because it serves as a universal proxy for the objective function applicable to both open weights and black-box models (via API). This allows our analysis to remain generalizable across different model families.
>
>
> > Response to **Q3: Gradient Conflict Resolution**
>
> In our framework, gradient conflict, common in Multi-Task Learning, is less relevant to the *generation* of reasoning trajectories. In the RAML setup (and standard RL/SFT), each question acts as an independent task during the inner loop (inference). The "optimization" occurs via the forward pass of the trajectory. While different trajectories might represent different strategies (deductive vs. heuristic), they do not "conflict" in the backward pass sense during inference. Instead, diverse trajectories during training act as a robust support set, improving the meta-optimizer (the LLM parameters).
>
> > Response to **Q4: Efficient Reasoning & Early Stopping**
>
> We explore efficiency in **Section 4** (Incentivizing Reasoning Efficiency). We do not suggest arbitrary early stopping; rather, we propose that an *optimal* inner-loop path exists that is shorter than standard CoT. As shown in **Figure 12**, summarizing trajectories to remove redundant "optimization steps" maintains performance while reducing token count. Practically, we could monitor metrics like confidence or entropy stability to detect convergence of the inner loop and trigger an "End of Thinking" token. This approach cannot fully guarantee the complete consistency of answer generation, but it can serve as a **trade-off strategy between efficiency and accuracy**.
>
> ---
>
> Thank you for your insightful review. We hope this addresses your concerns and remain open to further discussion.

---

> ### Comment · Reviewer_PWxJ · 2025-11-27
> **Comments**
>
> Thanks for the authors for the detailed response.  Most of the questions are addressed. However:
>
> Q1. While STEM encompasses diverse scientific disciplines including biology, chemistry, and physics, the number of examples are limited.
>
> Q2. While primary probe is applicable to both open weights and black-box models (via API). Most of the experiments in this paper were with the open weights models.
>
> Therefore, I will maintain my original score. Nonetheless, I am still in favor of accepting this paper.

---

### Official Review · Reviewer_dnUv · 2025-11-04

**Soundness:** 3
**Presentation:** 3
**Contribution:** 2
**Rating:** 6
**Confidence:** 3

**Summary:**

This paper proposes RAML, a meta-learning perspective on LLM reasoning. The core idea is to treat chain-of-thought trajectories as pseudo–gradient descent steps on an implicit loss surface, so that answering each question becomes an inner-loop adaptation process. The authors give a theoretical formulation using a simplified transformer block, argue that attention over reasoning tokens can be reparameterized as parameter updates, and visualize the resulting “loss landscapes”. On the empirical side, they study several factors through this lens: SFT vs. RL training, on-policy vs. off-policy trajectories, trajectory length and reflection tokens, and within-domain vs. cross-domain generalization. Finally, they propose two simple strategies inspired by meta-learning (more trajectories per question and summarized trajectories) and show that these can improve or preserve reasoning performance under different budget settings.

**Strengths:**

* The paper offers a clean and coherent framework that connects two active areas: long-CoT LLM reasoning and gradient-based meta-learning. The mapping between reasoning trajectories and inner-loop optimization steps is clear.

* Theoretical, empirical, and conceptual parts are relatively consistent with each other. The simplified transformer analysis, loss-landscape visualizations, and behavioral experiments all support the same narrative.

* The experimental section is fairly extensive: multiple benchmarks (AIME24, MATH500-L5, LiveMathBench-Hard, GPQA, LiveCodeBench), several training recipes (SFT, Zero-GRPO, SFT+GRPO), and analyses on trajectory length, token types, and generalization.

* The writing and presentation are clear. Figures and tables are well organized and help the reader see the connection between the meta-learning view and concrete LLM behaviors.

**Weaknesses:**

Overall, this paper proposes an interesting angle to try to unify several recent lines of work in reasoning, and it argues for its story and research questions in a fairly consistent way from the theoretical, empirical, and expository perspectives. The paper reads as well-reasoned, and the logic is relatively self-contained and closed; these aspects are very nice.

However, my main concern is that the authors seem to build a new conceptual system to reinterpret existing problems from a different angle, and in the end almost everything is devoted to repeatedly emphasizing that this new system aligns very well with how people already think about today's LLM-reasoning tasks. The new framework, after all this effort, does not appear to bring many genuinely new directions or critiques that emerge from “re-thinking” the space. Most of the conclusions and proposed future directions look quite similar to what one would obtain from the original, non-meta-learning framing of reasoning.

Of course, this is also a positive sign in one sense: the authors do convincingly show that the two perspectives are compatible, and I am convinced on that point. But if, under both systems, most things neither conflict nor lead to any critical tension, then it is not yet clear what concrete difference it makes to adopt this new perspective. This gap between “nice unifying story” and “new actionable or disruptive insight” is my main reservation about the contribution.

**Questions:**

* If most findings, intuitions, and future directions remain essentially the same under both today's LLM-reasoning perspective (without the meta-learning concept) and the proposed meta-learning perspective, what is the practical benefit of switching to or adopting this new framework? How should a practitioner or researcher reason differently about model design or training because of RAML?

* During the rebuttal, I am especially curious to see whether the authors can use the meta-learning angle to surface **different** critical viewpoints or guidance for several of the current major directions in reasoning, rather than future guidance that is almost completely aligned with existing narratives. If RAML can highlight specific tensions, trade-offs, or failure modes that the standard view misses, that would help the paper go beyond the feeling of merely being “self-consistent” and would make its contribution more instructive and potentially disruptive for the community.

---

> ### Author Response · Authors · 2025-11-21
> **Response to Reviewer dnUv**
>
> Dear reviewer,
>
> We sincerely appreciate your recognition of our work’s clean framework, extensive experiments, and clear presentation. We value your thoughtful questions regarding the practical utility of the RAML framework. Please find our responses below.
>
> > Response to Weakness & Q1 (Practical Benefit):
>
> The primary motivation for integrating LLM reasoning with meta-learning is twofold:
>
> - **Interpretation:** We leverage the established theoretical frameworks of meta-learning to decipher *why* certain phenomena occur in LLM reasoning. For instance, interpreting reasoning trajectories as inner-loop optimization steps explains why longer Chain-of-Thought (CoT) sequences often yield better performance (more optimization steps) and why SFT stabilizes reasoning (learning from an "oracle" optimizer).
> - **Advancement:** Beyond interpretation, RAML provides a structured methodology to derive new strategies for improving LLM reasoning. By mapping reasoning concepts to meta-learning mechanics (e.g., "trajectories per question" $\leftrightarrow$ "support set size"), we can port successful meta-learning techniques into the reasoning domain.
>
> > Response to Q2 (Novel Guidance vs. Existing Narratives):
>
> We agree that offering critical, divergent guidance is essential. RAML does indeed highlight trade-offs that standard views might miss. For example, the data scaling strategies discussed in **Section 4** and **Appendix I.1**.
> Current community consensus for SFT focuses largely on increasing the diversity of unique questions ($|\mathcal{Q}|$). However, the RAML perspective suggests that increasing the **support set size,** i.e., the number of distinct reasoning trajectories per question ($|\mathcal{T}_i|$), is equally critical for stabilizing the inner-loop optimization.
>
> - **New Insight:** Our experiments (Figure 7 and Figure 9) empirically validate that training with multiple trajectories per question significantly boosts performance.
> - **Implication:** This suggests a shift in data curation strategy: rather than solely maximizing question breadth, practitioners should also maximize trajectory density per question to improve the model's ability to generalize the "optimization" process. This is a specific, actionable insight derived directly from the meta-learning analogy that differs from standard "scale the dataset" narratives.
>
> ---
>
> Finally, thank you again for your detaile feedback. We welcome any further questions.

---

### Author Response · Authors · 2025-11-21
**General Response**

We sincerely thank all reviewers for their thorough and thoughtful reviews of our paper. We are grateful for the positive feedback on RAML’s clean framework, extensive experiments, and clear presentation, as well as the constructive suggestions provided. These have helped us significantly improve the clarity and rigor of our work. We provide a detailed response to each comment. Here we highlight our major clarifications regarding the theoretical framework, practical utility, and experimental methodology. We hope our responses can properly address your concerns.

- **Theoretical Grounding and Conceptual Feasibility (Reviewers dnUv, PWxJ, 3CaD, zmaQ):** We want to clarify that RAML serves primarily as a descriptive framework to interpret LLM behaviors rather than a prescriptive mathematical proof of convergence. Our goal is to establish a conceptual bridge between forward computation and pseudo-gradient updates to motivate the framework.
    - **Feasibility:** **Proposition 2.1** demonstrates the structural feasibility of this mapping within standard Transformer layers.
    - **Validation:** While quantifying exact approximation error is non-trivial due to the abstract objective , we provide strong empirical evidence (e.g., **Figures 2 and 13**) showing consistent alignment between our framework and observed model behaviors. We emphasize that explicit calculation of second-order gradients is unnecessary for this analogy to hold, as standard backpropagation in SFT/RL implicitly optimizes the "optimizer" (the LLM).
- **Practical Utility and Novel Data Strategies (Reviewers dnUv, 3CaD):** We highlight that RAML offers critical, divergent guidance compared to standard data scaling narratives.
    - **New Insight:** While current consensus emphasizes increasing question diversity ($|\mathcal{Q}|$), our meta-learning perspective suggests that increasing **trajectory density per question** (Support Set Size) is equally critical for stabilizing inner-loop optimization.
    - **Actionable Strategy:** Our experiments (**Figure 7 and Table 3**) validate that "few questions, multi-trajectory" datasets can yield greater gains. This provides a structured methodology to derive new strategies for improving LLM reasoning beyond simple interpretation.
- **Validity of Proxy Objectives and Metrics (Reviewers PWxJ, 3CaD, zmaQ):** We clarify our choice of metrics and address concerns regarding objective functions.
    - **Logits as Proxy:** Since reasoning is a discrete process with an intractable true objective, using the log-likelihood of the correct answer is a standard, principled proxy. We deliberately chose logits over internal state probes (like causal tracing) to ensure our analysis remains generalizable to both open-weights and closed-box models.
    - **Robustness:** To ensure statistical significance, we have incorporated explicit error bars and confidence intervals (via bootstrap resampling) in our revised results, averaging performance over multiple runs (e.g., Pass@8).

---

### Meta-Review · Area_Chair_msq8 · 2026-01-08

**Summary:**

Reviewers share the following major concerns:
1. While they appreciated the new perspective of understanding long CoT reasoning as a meta-learning process, reviewers challenged the unique insights brought by this perspective. In particular, reviewers commented that findings from the experiments have been found in the prior work, yet the new interpretation provided by this work is not convincing.
2. Reviewers challenged the rigor of the proposed meta-learning perspective (e.g., how exactly does the generative CoT process connect to the classic optimization process, how to quantify the equivalence, how does the second-order gradient of MAML happen in generative CoT without parameter updates, how to characterize the corresponding properties of MAML in CoT, etc.).

**Reviewer Concerns:**

For 1, the authors emphasized the performance gain from diversifying reasoning trajectories during training time, an idea inspired by the effect of increasing the support set in meta-learning.

For 2, the authors admitted the lack of rigor in their formulation and emphasized that their purpose of this work is to show the feasibility of building the connection between CoT reasoning and meta-learning.

While Concern 1 may be fine (though it will be more convincing if more than one application can be showcased), Concern 2 may not be considered addressed by the reviewers (especially Reviewers 3CaD and zmaQ). While the direction of formulating CoT reasoning as a meta-learning process is promising, I share the same impression with Reviewers 3CaD and zmaQ that the current formulation is a bit shallow and lacking sufficient rigor, which makes it less convincing.

**Reviewer Scores:**

Reviewer PWxJ (initial score 6) responded with their intent to keep the score unchanged. They also expressed their support for this paper.
However, I'm concerned that the two reviewers who gave an initial score of 2 and raised severe concerns about the lack of rigor of the formulation may not increase the score based on the authors' response.

---

### Decision · Program_Chairs · 2026-01-26

Reject